# Relative anomalies
# in (2+1)D symmetry enriched topological states

**Maissam Barkeshli[1] and Meng Cheng[2]**

**1** Department of Physics, Condensed Matter Theory Center, and Joint Quantum Institute,
University of Maryland, College Park, Maryland 20742, USA
**2** Department of Physics, Yale University, New Haven, CT 06511-8499, USA

## Abstract

Certain patterns of symmetry fractionalization in topologically ordered phases of matter are anomalous, in the sense that they can only occur at the surface of a higher dimensional symmetry-protected topological (SPT) state. An important question is to determine how to compute this anomaly, which means determining which SPT hosts a given symmetry-enriched topological order at its surface. While special cases are known, a general method to compute the anomaly has so far been lacking. In this paper we propose a general method to compute relative anomalies between different symmetry fractionalization classes of a given (2+1)D topological order. This method applies to all types of symmetry actions, including anyon-permuting symmetries and general space-time reflection symmetries. We demonstrate compatibility of the relative anomaly formula with previous results for diagnosing anomalies for $\mathbb{Z}_2^T$ space-time reflection symmetry (e.g. where time-reversal squares to the identity) and mixed anomalies for $U(1) \times \mathbb{Z}_2^T$ and $U(1) \rtimes \mathbb{Z}_2^T$ symmetries. We also study a number of additional examples, including cases where space-time reflection symmetries are intertwined in non-trivial ways with unitary symmetries, such as $\mathbb{Z}_4^T$ and mixed anomalies for $\mathbb{Z}_2 \times \mathbb{Z}_2^T$ symmetry, and unitary $\mathbb{Z}_2 \times \mathbb{Z}_2$ symmetry with non-trivial anyon permutations.

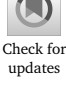

# 1  Introduction

The last few years in condensed matter physics have seen major progress in our understanding of symmetry and its interplay with topological degrees of freedom in gapped quantum many-body systems. In the context of quantum field theory, these results translate into progress regarding the characterization and classification of topological quantum field theories with symmetry.

In the absence of any symmetry, gapped quantum systems in two and higher spatial dimensions can still form different phases of matter, distinguished by their topological order [1–3]. In (2+1) space-time dimensions, it is believed that distinct gapped phases of quantum many-body systems can be fully characterized by a pair of objects, $(\mathcal{C}, c_-)$. $\mathcal{C}$ is a unitary modular tensor category, sometimes referred to as the algebraic theory of anyons, which describes the fusion and braiding properties of anyons, which are topologically non-trivial finite-energy quasiparticle excitations [2,4,5]. $c_-$ is the chiral central charge, and dictates the low temperature specific heat and thermal conductivity on the boundary of the system. In a system where the microscopic constituents are all bosons, $\mathcal{C}$ determines $c_-$ modulo 8, while in a system of fermions $c_-$ is determined by $\mathcal{C}$ modulo $1/2$.

When $\mathcal{C}$ is trivial, the system forms an invertible state. This means that the many-body ground state possesses an inverse state, such that the original state and its inverse together can be adiabatically converted to a trivial product state without closing the bulk energy gap. In the presence of a symmetry group $G$, an important class of invertible states are referred to as symmetry-protected topological (SPT) states [6]. These are states that can be adiabatically connected to a trivial product state if the symmetry can be broken, but not while preserving the symmetry. A wide class of $d$-dimensional SPT states can be classified using topological effective actions associated with the group $G$, which results in the group cohomology classification $\mathcal{H}^{d+1}[G, U(1)]$ [7,8]. More generally, sophisticated mathematical theories have been developed to classify invertible and SPT states in general dimensions in terms of generalized cohomology theories [9–12].

When the anyon theory $\mathcal{C}$ is non-trivial, the resulting quantum many-body ground state is non-trivial even in the absence of any symmetry as it is not possible to adiabatically transform the ground state to a trivial product state without closing the bulk energy gap. In the presence of a symmetry group $G$, some topologically ordered states may be disallowed, while others split into distinct symmetry-enriched topological (SETs) phases. Different SETs described by the same topological order $(\mathcal{C}, c_-)$ cannot be adiabatically connected to each other while preserving the symmetry, although they can be adiabatically connected if the symmetry is broken along the path.

A hallmark of SETs is that the topologically non-trivial excitations can carry fractional quantum numbers of the symmetry. Well-known examples include the fractional electric charge carried by quasiparticles in fractional quantum Hall states, or the neutral spin-1/2 "spinon" excitations in quantum spin liquids. The different patterns of symmetry fractionalization partially distinguish SETs with the same topological order $(\mathcal{C}, c_-)$. In the past few years, it was understood that symmetry fractionalization can also be classified in terms of group cohomology [13,14].

In addition to the symmetry fractionalization patterns, different SETs are characterized by different fusion and braiding properties of symmetry defects that can be introduced into the system. Ref. [14] developed a general algebraic theory of symmetry defects for unitary, space-time orientation preserving symmetries, known as a $G$-crossed braided tensor category theory (see also [15,16] for related work). The $G$-crossed braided tensor category is expected to fully characterize the distinction between different SETs in (2+1) dimensions. Concretely, it consists of a set of data $\{\rho, N, F, R, U, \eta\}$, subject to a number of consistency conditions and

gauge equivalences.

An intriguing property of symmetry fractionalization is that it is possible for some symmetry fractionalization classes to be anomalous, in the sense that the given symmetry fractionalization class cannot occur in purely (2+1) dimensions, but can occur at the (2+1)D boundary of a (3+1)D SPT state [17]. In the language of high energy theory, the associated topological field theory has a 't Hooft anomaly. A simple example is the case of a $\mathbb{Z}_2$ spin liquid with $\mathbb{Z}_2^{\mathbf{T}}$ time-reversal symmetry (with time-reversal $\mathbf{T}^2 = \mathbb{1}$), where the $\mathbb{Z}_2$ charge and flux both carry a Kramers degeneracy [17]. A basic question, then, is to determine which (3+1)D SPT state is required to host a given SET at its (2+1)D surface. We refer to this as "computing the symmetry fractionalization anomaly," or simply "computing the anomaly" associated with a given SET. Since almost all SPTs in (3+1) dimensions fall within the group cohomology classification $\mathcal{H}^4[G, U(1)]$, it follows that with a few exceptions, computing the anomaly amounts to computing an element of $\mathcal{H}^4[G, U(1)]$ given the data that describes a given symmetry fractionalization class.

In general, the problem of computing anomalies for SETs has not been fully solved except for certain special cases. For cases where the symmetry is unitary, space-time orientation preserving, and does not permute anyon types, a formula for the $\mathcal{H}^4$ anomaly was presented in Refs. [14,18]. Ref. [14] computed the anomaly by explicitly solving the $G$-crossed consistency equations up to an $\mathcal{H}^4$ co-cycle. Ref. [18] computed the anomaly in a different manner by following a derivation of an obstruction formula for group extensions of fusion categories developed by Etingof, Nikshych, and Ostrik [19]. However these formulas were neither generalized to the case where symmetries may permute anyons nor to symmetries that involve space-time reflections. For Abelian topological orders with Abelian (unitary orientation-preserving) symmetry groups that do not permute quasiparticle types, a bulk-boundary correspondence was also developed in Ref. [20].

For the case of space-time reflection symmetries where time-reversal or spatial reflection square to the identity (referred to as $\mathbb{Z}_2^{\mathbf{T}}$ and $\mathbb{Z}_2^{\mathbf{r}}$ respectively), methods to compute the anomaly were subsequently derived in Ref. [21] for bosonic systems, in Ref. [22,23] for fermionic systems, and independently conjectured for both in Ref. [24]. Ref. [25] also subsequently developed an alternate derivation. Ref. [26] has also recently extended the derivation of Ref. [21] to fermionic systems.

Recently, anomaly indicators for $U(1) \times \mathbb{Z}_2^{\mathbf{T}}$ and $U(1) \rtimes \mathbb{Z}_2^{\mathbf{T}}$ were also derived in Ref. [27], allowing a general computation of mixed anomalies between $U(1)$ and $\mathbb{Z}_2^{\mathbf{T}}$ symmetry. For general symmetry groups that involve space-time reflections, the only known result is an anomaly formula for Abelian toric code topological order when anyons are not permuted by symmetries, derived from an explicit microscopic construction of SET phases in Ref. [28].

Despite the above progress, to date a general method has not been developed for computing symmetry fractionalization anomalies in cases where symmetries permute anyon types, or where time-reversal or spatial reflection symmetry are intertwined in non-trivial ways with unitary orientation-preserving symmetries. One may consider, for example, systems with space-time reflection symmetries where time-reversal does not square to the identity, but rather squares to a non-trivial unitary symmetry, corresponding to the group $\mathbb{Z}_4^{\mathbf{T}}$. This would physically arise if the underlying microscopic time-reversal symmetry is not a true symmetry of the system, but rather time-reversal combined with a unitary symmetry, such as a spin rotation, is a symmetry. Then the effective time-reversal symmetry may not square to the identity but rather to a unitary spin rotation symmetry. On the other hand, in cases such as $G \times \mathbb{Z}_2^{\mathbf{T}}$ symmetry, where $G$ is a unitary space-time orientation preserving symmetry, there may be mixed anomalies which we currently do not know how to compute except in the special case where $G = U(1)$.

An important observation is that while symmetry fractionalization itself is in general char-

acterized by a complicated set of data [14], the difference between symmetry fractionalization classes forms an Abelian group, classified by $\mathcal{H}^2_{[\rho]}[G, \mathcal{A}]$ [14]. Here $\rho$ determines how the symmetries permute anyons, while $\mathcal{A}$ is an Abelian group that arises from the group structure of fusion of Abelian anyons. In other words, after fixing the way symmetries permute anyons by fixing $\rho$, two different symmetry fractionalization classes for a given topological order can then be related to each other by an element $[\mathbf{t}] \in \mathcal{H}^2_{[\rho]}[G, \mathcal{A}]$. Therefore, $[\mathbf{t}]$ should allow us to specify the *relative anomaly*, that is the difference in anomalies, between two symmetry fractionalization classes. Note that since anomalies, or equivalently SPTs in one higher dimension, form an Abelian group, the difference in anomalies is a well-defined notion.

In this paper, we explain how to compute relative anomalies for general symmetries, including symmetries that permute anyon types and also general space-time reflection symmetries. In the case of unitary space-time orientation preserving symmetries, the anomaly formula we derive, presented in Eq. (44), was previously also derived as an obstruction to gauging in Ref. [29] through a more abstract mathematical formalism. However the logical arguments leading to our derivation and its interpretation are somewhat different from that of Ref. [29]. Moreover, the mathematical structure and equations that we use are entirely in terms of the data and consistency conditions presented in Ref. [14].

To treat space-time reflection symmetries, we propose a method to generalize the G-crossed braided tensor category equations presented in Ref. [14] to characterize space-time reflection defects. Our proposal proceeds by incorporating additional labels to keep track of local space-time orientations. This allows us to extend the derivation of the relative anomaly formula to the case of general space-time reflection symmetries, which results in some minor modifications to the unitary space-time orientation preserving case (see Eq. (51)).

We subsequently use the relative anomaly formula to reproduce previous results for $\mathbb{Z}_2^{\mathbf{T}}$, $U(1) \times \mathbb{Z}_2^{\mathbf{T}}$, and $U(1) \rtimes \mathbb{Z}_2^{\mathbf{T}}$ anomalies [21, 27] in a completely different way, thus providing a highly non-trivial consistency check on the correctness of this approach.

We then use the relative anomaly formula to study a variety of previously inaccessible examples involving more general time-reversal symmetries. These include anomalies for $\mathbb{Z}_4^{\mathbf{T}}$ symmetry and mixed anomalies for $\mathbb{Z}_2 \times \mathbb{Z}_2^{\mathbf{T}}$ symmetries. A few notable simple examples are as follows. We find that the $\mathbb{Z}_2$ spin liquid (toric code) state with $\mathbb{Z}_4^{\mathbf{T}}$ symmetry is anomalous when the electric and magnetic particles carry half charge under $\mathbf{T}^2$. This provides a generalization of the anomalous eTmT state [17, 30] to the case of $\mathbb{Z}_4^{\mathbf{T}}$ symmetry, which we dub the anomalous $\mathrm{eT}^2\mathrm{mT}^2$ state. By studying $\mathbb{Z}_2 \times \mathbb{Z}_2^{\mathbf{T}}$ symmetry for the $\mathbb{Z}_2$ spin liquid (toric code) state, we find a host of novel types of mixed anomalies, summarized for the case of no permutations in Table 2.

We also explicitly study novel examples of anomalous symmetry fractionalization classes for unitary $\mathbb{Z}_2 \times \mathbb{Z}_2$ symmetry where anyons are permuted. This, for example, leads us to find anomalies associated with charge-conjugation symmetry in $U(1)_N$ Chern-Simons theories, summarized in Table 1.

Our analysis also yields as a byproduct new invariants for $\mathbb{Z}_2$ symmetry fractionalization, $\lambda_a, \tilde{\lambda}_a = \pm 1$ (see Eq. (106), (107)), which do not require $a$ to be self-dual. $\lambda_a = \pm 1$ is defined whenever $a$ is invariant under the action of the $\mathbb{Z}_2$ symmetry and determines whether $a$ carries integer or fractional charge under $\mathbb{Z}_2$. $\tilde{\lambda}_a$ is defined whenever $a$ is permuted to its topological charge conjugate under the action of the $\mathbb{Z}_2$ symmetry and changes by $\pm 1$ when the $\mathbb{Z}_2$ symmetry fractionalization class is changed.

## 2  Symmetry fractionalization

### 2.1  Review of UMTC notation

Here we briefly review the notation that we use to describe UMTCs. For a more comprehensive review of the notation that we use, see e.g. Ref. [14]. The topologically non-trivial quasiparticles of a (2+1)D topologically ordered state are equivalently referred to as anyons, topological charges, and quasiparticles. In the category theory terminology, they correspond to isomorphism classes of simple objects of the UMTC.

A UMTC $\mathcal{C}$ contains splitting spaces $V_c^{ab}$, and their dual fusion spaces, $V_{ab}^c$, where $a, b, c \in \mathcal{C}$ are the anyons. These spaces have dimension $\dim V_c^{ab} = \dim V_{ab}^c = N_{ab}^c$, where $N_{ab}^c$ are referred to as the fusion rules. They are depicted graphically as:

$$(d_c/d_a d_b)^{1/4} \quad \vcenter{\hbox{\includegraphics{}}} \quad = \langle a, b; c, \mu | \in V_{ab}^c, \tag{1}$$

$$(d_c/d_a d_b)^{1/4} \quad \vcenter{\hbox{\includegraphics{}}} \quad = |a, b; c, \mu \rangle \in V_c^{ab}, \tag{2}$$

where $\mu = 1, \ldots, N_{ab}^c$, $d_a$ is the quantum dimension of $a$, and the factors $\left(\frac{d_c}{d_a d_b}\right)^{1/4}$ are a normalization convention for the diagrams.

We denote $\bar{a}$ as the topological charge conjugate of $a$, for which $N_{a\bar{a}}^1 = 1$, i.e.

$$a \times \bar{a} = 1 + \cdots \tag{3}$$

Here 1 refers to the identity particle, i.e. the vacuum topological sector, which physically describes all local, topologically trivial excitations.

The $F$-symbols are defined as the following basis transformation between the splitting spaces of 4 anyons:

$$\vcenter{\hbox{\includegraphics{}}} = \sum_{f,\mu,\nu} \left[F_d^{abc}\right]_{(e,\alpha,\beta)(f,\mu,\nu)} \vcenter{\hbox{\includegraphics{}}}. \tag{4}$$

To describe topological phases, these are required to be unitary transformations, i.e.

$$\left[\left(F_d^{abc}\right)^{-1}\right]_{(f,\mu,\nu)(e,\alpha,\beta)} = \left[\left(F_d^{abc}\right)^{\dagger}\right]_{(f,\mu,\nu)(e,\alpha,\beta)} = \left[F_d^{abc}\right]^*_{(e,\alpha,\beta)(f,\mu,\nu)}. \tag{5}$$

The $R$-symbols define the braiding properties of the anyons, and are defined via the the following diagram:

$$\vcenter{\hbox{\includegraphics{}}} = \sum_{\nu} \left[R_c^{ab}\right]_{\mu\nu} \vcenter{\hbox{\includegraphics{}}}. \tag{6}$$

Under a basis transformation, $\Gamma_c^{ab} : V_c^{ab} \to V_c^{ab}$, the $F$ and $R$ symbols change:

$$F_d^{abc} \to \tilde{F}_d^{abc} = \Gamma_e^{ab} \Gamma_d^{ec} F_d^{abc} [\Gamma_f^{bc}]^{\dagger} [\Gamma_d^{af}]^{\dagger},$$
$$R_c^{ab} \to \tilde{R}_c^{ab} = \Gamma_c^{ba} R_c^{ab} [\Gamma_c^{ab}]^{\dagger}. \tag{7}$$

These basis transformations are referred to as vertex basis gauge transformations. Physical quantities correspond to gauge-invariant combinations of the data.

The topological twist $\theta_a = e^{2\pi i h_a}$, with $h_a$ the topological spin, is defined via the diagram:

$$\theta_a = \theta_{\bar{a}} = \sum_{c,\mu} \frac{d_c}{d_a} \left[ R_c^{aa} \right]_{\mu\mu} = \frac{1}{d_a} \quad \underset{a}{\infty} \quad . \tag{8}$$

Finally, the modular, or topological, $S$-matrix, is defined as

$$S_{ab} = \mathcal{D}^{-1} \sum_c N_{\bar{a}b}^c \frac{\theta_c}{\theta_a \theta_b} d_c = \frac{1}{\mathcal{D}} \quad a \, b \quad , \tag{9}$$

where $\mathcal{D} = \sqrt{\sum_a d_a^2}$.

A quantity that we make extensive use of is the double braid, which is a phase if either $a$ or $b$ is an Abelian anyon:

$$\begin{array}{c} a \quad b \\ \Big\langle \Big\rangle \end{array} = M_{ab} \quad \begin{array}{c} a \quad b \\ \Big\uparrow \Big\uparrow \end{array} . \tag{10}$$

## 2.2 Topological symmetry and braided auto-equivalence

An important property of a UMTC $\mathcal{C}$ is the group of "topological symmetries," which are related to "braided auto-equivalences" in the mathematical literature. They are associated with the symmetries of the emergent TQFT described by $\mathcal{C}$, irrespective of any microscopic global symmetries of a quantum system in which the TQFT emerges as the long wavelength description.

The topological symmetries consist of the invertible maps

$$\varphi : \mathcal{C} \to \mathcal{C}. \tag{11}$$

The different $\varphi$, modulo equivalences known as natural isomorphisms, form a group, which we denote as $\mathrm{Aut}(\mathcal{C})$. [14]

The symmetry maps can be classified according to a $\mathbb{Z}_2 \times \mathbb{Z}_2$ grading, defined by

$$q(\varphi) = \begin{cases} 0 & \text{if } \varphi \text{ is not time-reversing} \\ 1 & \text{if } \varphi \text{ is time-reversing,} \end{cases} \tag{12}$$

$$p(\varphi) = \begin{cases} 0 & \text{if } \varphi \text{ is spatial parity even} \\ 1 & \text{if } \varphi \text{ is spatial parity odd.} \end{cases} \tag{13}$$

Here time-reversing transformations are anti-unitary, while spatial parity odd transformations involve an odd number of reflections in space, thus changing the orientation of space. Thus the topological symmetry group can be decomposed as

$$\mathrm{Aut}(\mathcal{C}) = \bigsqcup_{q,p=0,1} \mathrm{Aut}_{q,p}(\mathcal{C}). \tag{14}$$

$\mathrm{Aut}_{0,0}(\mathcal{C})$ is therefore the subgroup corresponding to topological symmetries that are unitary and space-time parity even (this is referred to in the mathematical literature as the group of "braided auto-equivalences"). The generalization involving reflection and time-reversal symmetries appears to be beyond what has been considered in the mathematics literature to date.

It is also convenient to define

$$\sigma(\varphi) = \begin{cases} 1 & \text{if } \varphi \text{ is space-time parity even} \\ * & \text{if } \varphi \text{ is space-time parity odd.} \end{cases} \tag{15}$$

A map $\varphi$ is space-time parity odd if $(q(\varphi) + p(\varphi))\ \mathrm{mod}\ 2 = 1$, and otherwise it is space-time parity even.

The maps $\varphi$ may permute the topological charges:

$$\varphi(a) = a' \in \mathcal{C}, \tag{16}$$

subject to the constraint that

$$
\begin{aligned}
N_{a'b'}^{c'} &= N_{ab}^{c}, \\
S_{a'b'} &= S_{ab}^{\sigma(\varphi)}, \\
\theta_{a'} &= \theta_{a}^{\sigma(\varphi)}.
\end{aligned}
\tag{17}
$$

The maps $\varphi$ have a corresponding action on the $F$- and $R$- symbols of the theory, as well as on the fusion and splitting spaces, which we will discuss in the subsequent section.

## 2.3 Global symmetry

Let us now suppose that we are interested in a system with a global symmetry group $G$. For example, we may be interested in a given microscopic Hamiltonian that has a global symmetry group $G$, whose ground state preserves $G$, and whose anyonic excitations are algebraically described by $\mathcal{C}$. The global symmetry acts on the topological quasiparticles and the topological state space through the action of a group homomorphism

$$[\rho] : G \to \mathrm{Aut}(\mathcal{C}). \tag{18}$$

We use the notation $[\rho_{\mathbf{g}}] \in \mathrm{Aut}(\mathcal{C})$ for a specific element $\mathbf{g} \in G$. The square brackets indicate the equivalence class of symmetry maps related by natural isomorphisms, which we define below. $\rho_{\mathbf{g}}$ is thus a representative symmetry map of the equivalence class $[\rho_{\mathbf{g}}]$. We use the notation

$$^{\mathbf{g}}a \equiv \rho_{\mathbf{g}}(a). \tag{19}$$

We associate gradings $q(\mathbf{g})$ and $p(\mathbf{g})$ by defining

$$
\begin{aligned}
q(\mathbf{g}) &\equiv q(\rho_{\mathbf{g}}), \\
p(\mathbf{g}) &\equiv p(\rho_{\mathbf{g}}), \\
\sigma(\mathbf{g}) &\equiv \sigma(\rho_{\mathbf{g}}).
\end{aligned}
\tag{20}
$$

In this section we consider the case with no spatial reflections, i.e. $p(\mathbf{g}) = 0$. The case with spatial reflection is discussed in Sec. 5.

$\rho_{\mathbf{g}}$ has an action on the fusion/splitting spaces:

$$\rho_{\mathbf{g}} : V_{ab}^{c} \to V_{^{\mathbf{g}}a\,^{\mathbf{g}}b}^{^{\mathbf{g}}c}. \tag{21}$$

This map is unitary if $q(\mathbf{g}) = 0$ and anti-unitary if $q(\mathbf{g}) = 1$. We write this as

$$\rho_{\mathbf{g}}|a, b; c, \mu\rangle = \sum_{\nu} [U_{\mathbf{g}}(^{\mathbf{g}}a, ^{\mathbf{g}}b; ^{\mathbf{g}}c)]_{\mu\nu} K^{q(\mathbf{g})} |^{\mathbf{g}}a, ^{\mathbf{g}}b; ^{\mathbf{g}}c, \nu\rangle, \tag{22}$$

where $U_{\mathbf{g}}(^{\mathbf{g}}a, ^{\mathbf{g}}b; ^{\mathbf{g}}c)$ is a $N_{ab}^{c} \times N_{ab}^{c}$ matrix, and $K$ denotes complex conjugation.

Under the map $\rho_{\mathbf{g}}$, the $F$ and $R$ symbols transform as well:

$$\rho_{\mathbf{g}}[F_{def}^{abc}] = U_{\mathbf{g}}(^{\mathbf{g}}a, ^{\mathbf{g}}b; ^{\mathbf{g}}e) U_{\mathbf{g}}(^{\mathbf{g}}e, ^{\mathbf{g}}c; ^{\mathbf{g}}d) F_{^{\mathbf{g}}d\,^{\mathbf{g}}e\,^{\mathbf{g}}f}^{^{\mathbf{g}}a\,^{\mathbf{g}}b\,^{\mathbf{g}}c} U_{\mathbf{g}}^{-1}(^{\mathbf{g}}b, ^{\mathbf{g}}c; ^{\mathbf{g}}f) U_{\mathbf{g}}^{-1}(^{\mathbf{g}}a, ^{\mathbf{g}}f; ^{\mathbf{g}}d) \tag{23}$$

$$= K^{q(\mathbf{g})} F_{def}^{abc} K^{q(\mathbf{g})}$$

$$\rho_{\mathbf{g}}[R_{c}^{ab}] = U_{\mathbf{g}}(^{\mathbf{g}}b, ^{\mathbf{g}}a; ^{\mathbf{g}}c) R_{^{\mathbf{g}}c}^{^{\mathbf{g}}a\,^{\mathbf{g}}b} U_{\mathbf{g}}(^{\mathbf{g}}a, ^{\mathbf{g}}b; ^{\mathbf{g}}c)^{-1} = K^{q(\mathbf{g})} R_{c}^{ab} K^{q(\mathbf{g})}, \tag{24}$$

where we have suppressed the additional indices that appear when $N^c_{ab} > 1$.

Importantly, we have

$$\kappa_{\mathbf{g},\mathbf{h}} \circ \rho_{\mathbf{g}} \circ \rho_{\mathbf{h}} = \rho_{\mathbf{gh}}, \tag{25}$$

where the action of $\kappa_{\mathbf{g},\mathbf{h}}$ on the fusion / splitting spaces is defined as

$$\kappa_{\mathbf{g},\mathbf{h}}(|a,b;c,\mu\rangle) = \sum_{\nu} [\kappa_{\mathbf{g},\mathbf{h}}(a,b;c)]_{\mu\nu} |a,b;c,\nu\rangle. \tag{26}$$

The above definitions imply that

$$\kappa_{\mathbf{g},\mathbf{h}}(a,b;c) = U_{\mathbf{g}}(a,b;c)^{-1} K^{q(\mathbf{g})} U_{\mathbf{h}}(\bar{^{\mathbf{g}}}a, \bar{^{\mathbf{g}}}b; \bar{^{\mathbf{g}}}c)^{-1} K^{q(\mathbf{q})} U_{\mathbf{gh}}(a,b;c), \tag{27}$$

where $\bar{\mathbf{g}} \equiv \mathbf{g}^{-1}$. $\kappa_{\mathbf{g},\mathbf{h}}$ is a natural isomorphism, which means that by definition,

$$[\kappa_{\mathbf{g},\mathbf{h}}(a,b;c)]_{\mu\nu} = \delta_{\mu\nu} \frac{\beta_a(\mathbf{g},\mathbf{h})\beta_b(\mathbf{g},\mathbf{h})}{\beta_c(\mathbf{g},\mathbf{h})}, \tag{28}$$

where $\beta_a(\mathbf{g},\mathbf{h})$ are U(1) phases.

## 2.4 Symmetry localization and fractionalization

Now let us consider the action of a symmetry $\mathbf{g} \in G$ on the full quantum many-body state of the system. Let $R_{\mathbf{g}}$ be the representation of $\mathbf{g}$ acting on the full Hilbert space of the theory. We consider a state $|\Psi_{a_1,\cdots,a_n}\rangle$ in the full Hilbert space of the system, which consists of $n$ anyons, $a_1, \cdots a_n$, at well-separated locations, which collectively fuse to the identity topological sector. Since the ground state is $G$-symmetric, we expect that the symmetry action $R_{\mathbf{g}}$ on this state possesses a property that we refer to as symmetry localization. This is the property that the symmetry action $R_{\mathbf{g}}$ decomposes as

$$R_{\mathbf{g}} |\Psi_{a_1,\cdots,a_n}\rangle \approx \prod_{j=1}^{n} U_{\mathbf{g}}^{(j)} U_{\mathbf{g}}(^{\mathbf{g}}a_1, \cdots, {}^{\mathbf{g}}a_n; 1) |\Psi_{{}^{\mathbf{g}}a_1,\cdots,{}^{\mathbf{g}}a_n}\rangle. \tag{29}$$

Here, $U_{\mathbf{g}}^{(j)}$ are unitary matrices that have support in a region (of length scale set by the correlation length) localized to the anyon $a_j$. The map $U_{\mathbf{g}}(^{\mathbf{g}}a_1, \cdots, {}^{\mathbf{g}}a_n; 1)$ is the generalization of $U_{\mathbf{g}}(^{\mathbf{g}}a, {}^{\mathbf{g}}b; {}^{\mathbf{g}}c)$, defined above, to the case with $n$ anyons fusing to vacuum. $U_{\mathbf{g}}(^{\mathbf{g}}a_1, \cdots, {}^{\mathbf{g}}a_n; 1)$ only depends on the global topological sector of the system – that is, on the precise fusion tree that defines the topological state – and not on any other details of the state, in contrast to the local operators $U_{\mathbf{g}}^{(j)}$. The $\approx$ means that the equation is true up to corrections that are exponentially small in the size of $U^{(j)}$ and the distance between the anyons, in units of the correlation length.

The choice of action $\rho$ defined above defines an element $[\mho] \in \mathcal{H}^3_{[\rho]}(G, \mathcal{A})$ [14]. If $[\mho]$ is non-trivial, then there is an obstruction to Eq. (29) being consistent when considering the associativity of three group elements. We refer to this as a symmetry localization anomaly, or symmetry localization obstruction. See. Ref. [14,31,32] for examples.[1]

If $[\mho]$ is trivial, so that symmetry localization as described by Eq. (29) is well-defined, then it is possible to define a notion of symmetry fractionalization [14]. When the action of $\rho$ is trivial, this is particularly simple to review. In this case, one can fix a gauge where

---

[1]A non-trivial obstruction $[\mho] \in \mathcal{H}^3_{[\rho]}(G, \mathcal{A})$ can be alternatively interpreted as the associated TQFT possessing a non-trivial 2-group symmetry, consisting of the 0-form symmetry group $G$ and the 1-form symmetry group $\mathcal{A}$, with $[\mho]$ characterizing the 2-group [14,19,33].

$U_{\mathbf{g}}(a, b; c) = 1$. Symmetry fractionalization then corresponds to a possible choice of phase $\omega_a(\mathbf{g}, \mathbf{h})$:

$$U_{\mathbf{g}}^{(1)} U_{\mathbf{h}}^{(1)} = \omega_a(\mathbf{g}, \mathbf{h}) U_{\mathbf{gh}}^{(1)}. \tag{30}$$

One can show that $\omega_a \omega_b = \omega_c$ whenever $N_{ab}^c \neq 0$, which then implies that $\omega_a = M_{a\mathfrak{w}(\mathbf{g}, \mathbf{h})}$, where $\mathfrak{w}(\mathbf{g}, \mathbf{h}) \in \mathcal{A}$ is an Abelian anyon. One can show that associativity of three group elements requires that $\mathfrak{w}$ obey a 2-cocyle condition, while redefinitions of $U_{\mathbf{g}}^{(1)}$ allow one to change $\mathfrak{w}$ by a coboundary. It thus follows that the symmetry fractionalization pattern corresponds to an element $[\mathfrak{w}] \in \mathcal{H}^2(G, \mathcal{A})$.

When the action of $\rho$ is non-trivial, so that the anyons can be permuted by symmetries, the above analysis is more complicated. A detailed analysis [14] reveals that now symmetry fractionalization patterns are no longer characterized by group cohomology. Rather, different symmetry fractionalization classes can be related to each other by $[\mathbf{t}] \in \mathcal{H}_{[\rho]}^2(G, \mathcal{A})$. In mathematical parlance, symmetry fractionalization classes form an $\mathcal{H}_{[\rho]}^2(G, \mathcal{A})$ torsor.

In general, symmetry fractionalization is characterized by a consistent set of data $\{\eta\}$ and $\{U\}$, where $\{U\}$ was defined above. The data $\eta_a(\mathbf{g}, \mathbf{h})$ characterize the difference in phase obtained when acting "locally" on an anyon $a$ by $\mathbf{g}$ and $\mathbf{h}$ separately, as compared with $\mathbf{gh}$ (note $\eta_a(\mathbf{g}, \mathbf{h})$ is not the same as $\omega_a(\mathbf{g}, \mathbf{h})$ above). This can be captured through a physical process involving symmetry defects, as explained in the next section. There are two important consistency conditions for $U$ and $\eta$, which we will use repeatedly later in this paper [14]. The first one is

$$\frac{\eta_a(\mathbf{g}, \mathbf{h}) \eta_b(\mathbf{g}, \mathbf{h})}{\eta_c(\mathbf{g}, \mathbf{h})} = \kappa_{\mathbf{g}, \mathbf{h}}(a, b; c), \tag{31}$$

with $\kappa$ defined in terms of $U$ as in Eq. (27). The other one is

$$\eta_a(\mathbf{g}, \mathbf{h}) \eta_a(\mathbf{gh}, \mathbf{k}) = \eta_a(\mathbf{g}, \mathbf{hk}) \eta_{\bar{\rho}_{\mathbf{g}}^{-1}(a)}^{q(\mathbf{g})}(\mathbf{h}, \mathbf{k}). \tag{32}$$

These data are subject to an additional class of gauge transformations, referred to as symmetry action gauge transformations [14]:

$$U_{\mathbf{g}}(a, b; c) \rightarrow \frac{\gamma_a(\mathbf{g}) \gamma_b(\mathbf{g})}{\gamma_c(\mathbf{g})} U_{\mathbf{g}}(a, b; c),$$

$$\eta_a(\mathbf{g}, \mathbf{h}) \rightarrow \frac{\gamma_a(\mathbf{gh})}{(\gamma_{\bar{\mathbf{g}}_a}(\mathbf{g}))^{q(\mathbf{g})} \gamma_a(\mathbf{h})} \eta_a(\mathbf{g}, \mathbf{h}). \tag{33}$$

Recall $p(\mathbf{g}) = 0$ in the discussion of this section. We note that $U$ also changes under a vertex basis gauge transformation. Different gauge-inequivalent choices of $\{\eta\}$ and $\{U\}$ characterize distinct symmetry fractionalization classes [14].

# 3 Symmetry defects

One way to understand the classification of symmetry fractionalization is in terms of the properties of symmetry defects. A symmetry defect consists of a defect line in space, labeled by a group element $\mathbf{g} \in G$, which we sometimes refer to as a branch cut, and which can terminate at a point. In the (2+1)D space-time, the symmetry defect is thus associated with a two-dimensional branch sheet. A given branch cut line associated with $\mathbf{g}$ can have topologically distinct endpoints, which thus give rise to topologically distinct types of $\mathbf{g}$ defects; a particular

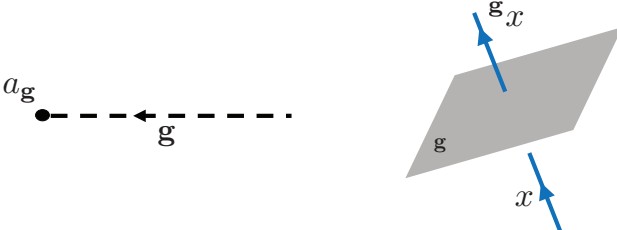

Figure 1: Left: A **g** defect in space. Branch cut is depicted by dashed line; there can be topologically distinct endpoints, labeled by $a_\mathbf{g}$. Right: **g** defect is a two-dimensional sheet in space-time; an anyon $x$ is permuted to $^\mathbf{g}x$ upon crossing the sheet.

topological class of **g** defect is thus labeled as $a_\mathbf{g}$. An anyon $x$ crossing the **g** defect branch cut is transformed into its permuted counterpart, $^\mathbf{g}x$ (see Fig. 1).

Here we focus on the case where the symmetry group is space-time orientation preserving and is represented unitarily on the quantum states. In this case, a complete algebraic theory of symmetry defects has been developed in [14], which captures the fusion and braiding properties of the symmetry defects. Without developing the full theory of symmetry defects, some elementary considerations can be used to reproduce the symmetry fractionalization classification, as we describe below.

First, we note that the defects can be organized into a $G$-graded fusion category,

$$\mathcal{C}_G = \bigoplus_{\mathbf{g} \in G} \mathcal{C}_\mathbf{g}, \tag{34}$$

where the simple objects of $\mathcal{C}_\mathbf{g}$ are the topologically distinct set of **g** defects. By considering states on a torus with a **g** defect wrapping one of the cycles, one can show that

$$|\mathcal{C}_\mathbf{g}| = |\mathcal{C}_0^\mathbf{g}|, \tag{35}$$

where $|\mathcal{C}_\mathbf{g}|$ is the number of topologically distinct **g** defects, and $|\mathcal{C}_0^\mathbf{g}|$ is the number of **g** invariant anyons.

## 3.1 Fusion rules of symmetry defects and relation to symmetry fractionalization

Fusion of the defects respects the group multiplication law associated with their branch cuts, so that

$$a_\mathbf{g} \times b_\mathbf{h} = \sum_{c_\mathbf{gh}} N_{ab}^c c_\mathbf{gh}. \tag{36}$$

For a given choice of fusion rules, one can consider a different state where the fusion rules are modified relative to those of the original state by allowing an anyon flux line associated with an anyon $\mathbf{t}(\mathbf{g}, \mathbf{h})$ to appear at the tri-junction between the branch sheets **g**, **h**, and **gh** (see Fig. 2). The fusion of the defects is thus modified to

$$a_\mathbf{g} \times b_\mathbf{h} = \mathbf{t}(\mathbf{g}, \mathbf{h}) \sum_c N_{ab}^c c_\mathbf{gh}. \tag{37}$$

Note that in our diagrammatic calculus, we pick the convention that the anyon line $\mathbf{t}(\mathbf{g}, \mathbf{h})$ propagates to the left in time.

Immediately, a number of important constraints appear for such a modification. The fusion of the branch sheets should be an invertible process, which requires that $\mathbf{t}(\mathbf{g}, \mathbf{h})$ be an Abelian

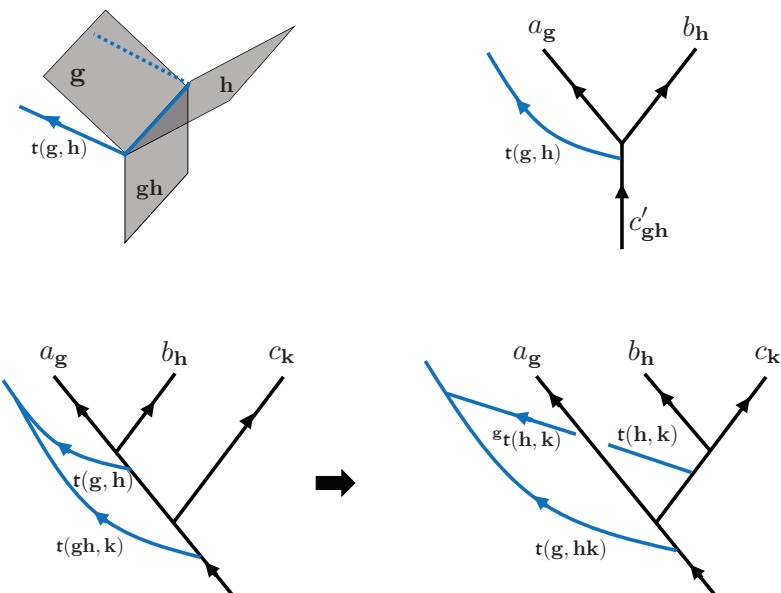

Figure 2: Top left: Fusion of defect branch sheets. Changing the symmetry fractionalization class by a 2-cocycle $t(\mathbf{g},\mathbf{h})$ changes the fusion of the defect worldsheets by the appearance of a Wilson line of $t(\mathbf{g},\mathbf{h})$ at the trijunction. Top right: Equivalent diagrammatic representation, where $a_\mathbf{g}$, $b_\mathbf{h}$ are the end-points of the $\mathbf{g}$ and $\mathbf{h}$ line defects. The fusion rule of the new theory thus becomes $a_\mathbf{g} \times b_\mathbf{h} = t(\mathbf{g},\mathbf{h}) \sum_{c_\mathbf{gh}} N^c_{ab} c_\mathbf{gh}$. We define $c'_\mathbf{gh} = t(\mathbf{g},\mathbf{h}) c_\mathbf{gh}$. Bottom panel: associativity of the defect fusion implies the 2-cocycle condition for $t(\mathbf{g},\mathbf{h})$.

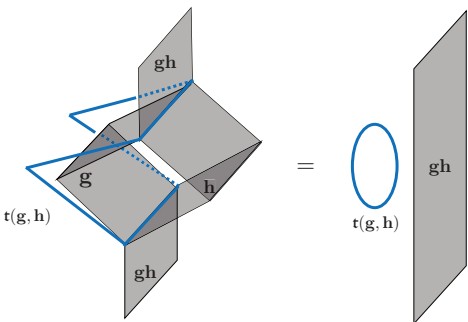

Figure 3: Splitting and fusing defect sheets $\mathbf{g}$ and $\mathbf{h}$ leaves behind an anyon loop for $t(\mathbf{g},\mathbf{h})$. Invertibility of the process thus requires $d_{t(\mathbf{g},\mathbf{h})} = 1$.

anyon. To see this, note that the process of fusing and then splitting the branch sheets to come back to the original configuration of branch sheets would leave behind an anyon loop associated with $t(\mathbf{g},\mathbf{h})$ (see Fig. 3). The anyon loop would give a quantum dimension factor $d_{t(\mathbf{g},\mathbf{h})}$ to the evaluation of defect diagrams; the invertibility of the process would then require $d_{t(\mathbf{g},\mathbf{h})} = 1$. We note that an alternative derivation of the fact that $t(\mathbf{g},\mathbf{h})$ is Abelian, starting from basic locality constraints on symmetry localization and symmetry fractionalization, is given in Ref. [14].

Furthermore, $t(\mathbf{g},\mathbf{h})$ must respect the associativity of fusion:

$$(a_\mathbf{g} \times b_\mathbf{h}) \times c_\mathbf{k} = a_\mathbf{g} \times (b_\mathbf{h} \times c_\mathbf{k}). \tag{38}$$

For the new fusion rules (37) to be associative, we are thus led to the constraint

$$t(\mathbf{g}, \mathbf{h})t(\mathbf{gh}, \mathbf{k}) = {}^{\mathbf{g}}t(\mathbf{h}, \mathbf{k})t(\mathbf{g}, \mathbf{hk}). \tag{39}$$

We can understand this equation diagrammatically as shown in Fig 2. Eq. (39) is the condition that $t(\mathbf{g}, \mathbf{h})$ be a 2-cocycle (twisted by the action of $\rho$). Since $t(\mathbf{g}, \mathbf{h})$ is itself ambiguous up to fusing $a_{\mathbf{g}}$ and $b_{\mathbf{h}}$ separately by an Abelian anyon, we thus also obtain an equivalence of $t(\mathbf{g}, \mathbf{h})$ under 2-coboundaries.

We see therefore that given a set of defect fusion rules, another set of defect fusion rules can be obtained given an element of $\mathcal{H}^2_{[\rho]}(G, \mathcal{A})$. In other words, the set of possible defect fusion rules forms an $\mathcal{H}^2_{[\rho]}(G, \mathcal{A})$ torsor.

The connection to symmetry fractionalization can be understood as follows. Symmetry fractionalization is characterized by the difference in phases obtained when acting 'locally' on an anyon by $\mathbf{g}$ and $\mathbf{h}$ separately, as compared with $\mathbf{gh}$. We can capture this by introducing the $\eta$ symbols, defined diagrammatically as follows:

$$\begin{array}{c}\includegraphics{eq40}\end{array} = \eta_x(\mathbf{g}, \mathbf{h}) \begin{array}{c}\includegraphics{eq40b}\end{array} . \tag{40}$$

The $U$ symbols, which define the action of symmetry group elements on the fusion and splitting spaces can also be defined diagrammatically, as shown:

$$\begin{array}{c}\includegraphics{eq41}\end{array} = \sum_{\nu} [U_{\mathbf{k}}(a, b; c)]_{\mu\nu} \begin{array}{c}\includegraphics{eq41b}\end{array} . \tag{41}$$

Different gauge-inequivalent choices of $\eta$ and $U$ define the notion of symmetry fractionalization [14]. When the trijunction of the defect branch sheets is modified to include an anyon flux line, we therefore see that $\eta_x(\mathbf{g}, \mathbf{h})$ changes, because the $x$ anyon line must be exchanged with the $t(\mathbf{g}, \mathbf{h})$ anyon line when being fully slid under the vertex, as shown in Fig. 4. This corresponds to the transformation $\eta_x(\mathbf{g}, \mathbf{h}) \rightarrow M_{xt(\mathbf{g}, \mathbf{h})}\eta_x(\mathbf{g}, \mathbf{h})$, where $M_{xt(\mathbf{g}, \mathbf{h})}$ is the phase obtained from a double braid between $x$ and $t(\mathbf{g}, \mathbf{h})$. This precisely corresponds to a change in the symmetry fractionalization class. In fact one can show that all possible symmetry fractionalization classes can be related to each other by such a change in the defect fusion rules [14], which implies that symmetry fractionalization classes are therefore related to each other by elements of $\mathcal{H}^2_{[\rho]}(G, \mathcal{A})$.

## 3.2 Relating $F$-symbols from different symmetry fractionalization classes

For unitary, orientation preserving symmetries, the defects form a $G$-graded fusion category. This means that in addition to the fusion rules (36), the defect fusion is characterized by $F$-symbols, which describe basis changes among different fusion trees for fusing three defects $a_{\mathbf{g}}$, $b_{\mathbf{h}}$, $c_{\mathbf{k}}$. For the defect theory to be anomaly-free, we require that the pentagon equation for the defect $F$-symbols be satisfied:

$$F^{fcd}_{egl}F^{abl}_{efk} = \sum_h F^{abc}_{gfh}F^{ahd}_{egk}F^{bcd}_{khl}, \tag{42}$$

where we have suppressed the group labels for ease of notation.

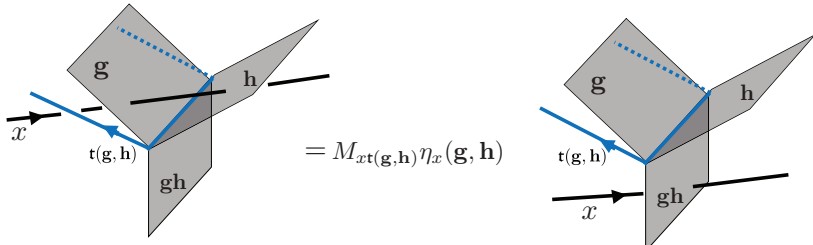

Figure 4: In the original theory, sliding an anyon line through the defect trijunction gave a phase $\eta_x(\mathbf{g}, \mathbf{h})$. In the new theory, the anyon line $x$ must also pass through the Abelian anyon line associated with $\mathbf{t}(\mathbf{g}, \mathbf{h})$, which picks up the mutual braiding phase $M_{x\mathbf{t}(\mathbf{g},\mathbf{h})}$ between $x$ and $\mathbf{t}(\mathbf{g}, \mathbf{h})$, as illustrated.

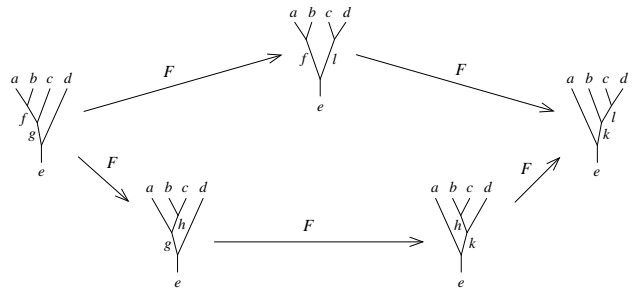

Figure 5: The Pentagon equation enforces the condition that different sequences of $F$-moves from the same starting fusion basis decomposition to the same ending decomposition gives the same result. Eq. (42) is obtained by imposing the condition that the above diagram commutes.

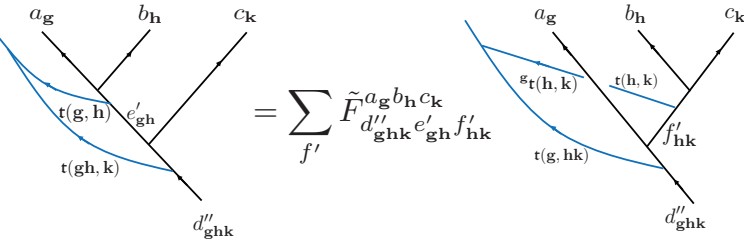

Figure 6: Diagram depicting $F$ symbols of the new theory, which we denote as $\tilde{F}$, in terms of labeling of objects in the original theory. We define $d''_{\mathbf{ghk}} = \mathbf{t}(\mathbf{g}, \mathbf{h})\mathbf{t}(\mathbf{gh}, \mathbf{k})d_{\mathbf{ghk}} = {}^{\mathbf{g}}\mathbf{t}(\mathbf{h}, \mathbf{k})\mathbf{t}(\mathbf{g}, \mathbf{hk})d_{\mathbf{ghk}}$, $e'_{\mathbf{gh}} = \mathbf{t}(\mathbf{g}, \mathbf{h})e_{\mathbf{gh}}$, $f'_{\mathbf{hk}} = \mathbf{t}(\mathbf{h}, \mathbf{k})f_{\mathbf{hk}}$.

Given two symmetry fractionalization classes related to each other by an element of $\mathcal{H}^2_{[\rho]}(G, \mathcal{A})$, the defect $F$ symbols must also change. Since the defect fusion rules change according to Eq. 37, we can also determine the $F$ symbols of the new theory given the data of the old theory.

Let us label the $F$ symbols of the new theory as $\tilde{F}$. $\tilde{F}$ is therefore associated with the diagram shown in Fig. 6. We can derive an explicit expression for $\tilde{F}$ by using the $F$ and $R$ moves of the original theory, as shown in Fig. 7. This leads to the following equation for $\tilde{F}$:

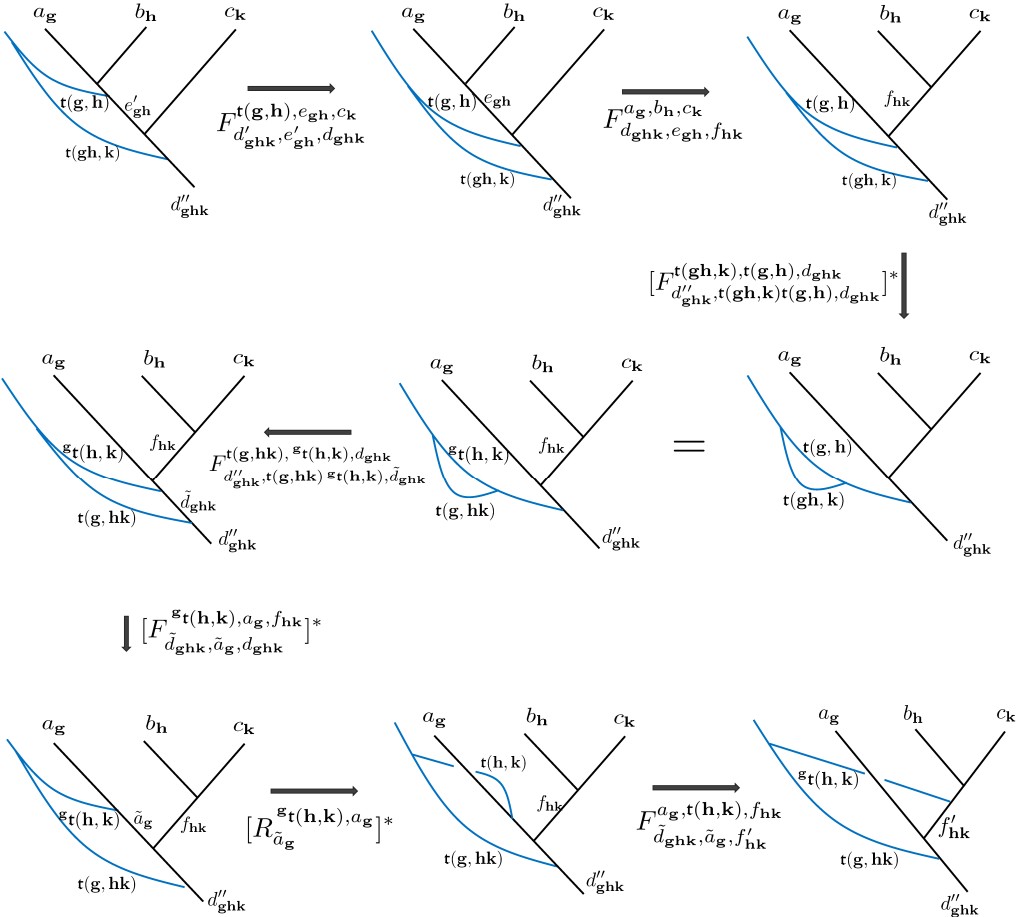

Figure 7: Sequence of moves in the original theory in order to determine the $F$ symbols, $\tilde{F}$, for defects in the new theory. The defect fusion rules in the new theory are twisted by an element $[\mathbf{t}] \in \mathcal{H}^2_{[\rho]}(G, \mathcal{A})$ relative to those of the original theory. We assume all defect and anyon lines have arrows pointing upwards.

.

$$\tilde{F}^{a_{\mathbf{g}},b_{\mathbf{h}},c_{\mathbf{k}}}_{d''_{\mathbf{ghk}},d'_{\mathbf{ghk}},f'_{\mathbf{hk}}} = F^{\mathbf{t(g,h)},e_{\mathbf{gh}},c_{\mathbf{k}}}_{d''_{\mathbf{ghk}},e'_{\mathbf{gh}},d_{\mathbf{ghk}}} F^{a_{\mathbf{g}},b_{\mathbf{h}},c_{\mathbf{k}}}_{d_{\mathbf{ghk}},e_{\mathbf{gh}},f_{\mathbf{hk}}} [F^{\mathbf{t(gh,k)},\mathbf{t(g,h)},d_{\mathbf{ghk}}}_{d''_{\mathbf{ghk}},\mathbf{t(gh,k)t(g,h)},d_{\mathbf{ghk}}}]^* F^{\mathbf{t(g,hk)},\,{}^{\mathbf{g}}\mathbf{t(h,k)},d_{\mathbf{ghk}}}_{d''_{\mathbf{ghk}},\mathbf{t(g,hk)}\,{}^{\mathbf{g}}\mathbf{t(h,k)},\tilde{d}_{\mathbf{ghk}}}$$

$$\times [F^{{}^{\mathbf{g}}\mathbf{t(h,k)},a_{\mathbf{g}},f_{\mathbf{hk}}}_{\tilde{d}_{\mathbf{ghk}},\tilde{a}_{\mathbf{g}},d_{\mathbf{ghk}}}]^* [R^{{}^{\mathbf{g}}\mathbf{t(h,k)},a_{\mathbf{g}}}_{\tilde{a}_{\mathbf{g}}}]^* F^{a_{\mathbf{g}},\mathbf{t(h,k)},f_{\mathbf{hk}}}_{\tilde{d}_{\mathbf{ghk}},\tilde{a}_{\mathbf{g}},f'_{\mathbf{hk}}} .$$

Note that we assume for ease of notation that all fusion coefficients are 0 or 1.

## 4 Relative anomaly calculation for symmetry fractionalization

In the previous sections we have discussed how one can consider two theories, where the "new" theory is related to the "original" theory by changing the defect fusion rules by an element $[\mathbf{t}] \in \mathcal{H}^2_{\rho}(G, \mathcal{A})$, which thus corresponds to a change in the symmetry fractionalization class. The relative anomaly derives from studying the consistency of the defect $F$ symbols for the new theory, denoted $\tilde{F}$ in the previous section, in relation to the consistency of the original

theory. In this section we use this consideration to derive a relative anomaly formula.

To do so, we consider the fusion of four defects in the original theory, together with Abelian anyons specified by $t : G \times G \to \mathcal{A}$, in order to recover the defect fusion of the new theory. This leads us to consider an analog of the pentagon equation for 4 defects, which now yields a non-trivial consistency condition obtained by following the 15 moves shown in Fig. 8.

Let us first assume that the original theory is fully consistent. In this case, following the 15 moves shown in Fig. 8 gives us a non-trivial equality:

$$\tilde{F}\tilde{F}\mathfrak{O}_r(\mathbf{g},\mathbf{h},\mathbf{k},\mathbf{l}) = \sum \tilde{F}\tilde{F}\tilde{F}, \tag{43}$$

where

$$\begin{aligned}
\mathfrak{O}_r(\mathbf{g},\mathbf{h},\mathbf{k},\mathbf{l}) =& R^{\,^{\mathbf{gh}}t(\mathbf{k},\mathbf{l}),t(\mathbf{g},\mathbf{h})}\,\eta_{\,^{\mathbf{gh}}t(\mathbf{k},\mathbf{l})}(\mathbf{g},\mathbf{h})[U_{\mathbf{g}}(\,^{\mathbf{g}}t(\mathbf{hk},\mathbf{l}),\,^{\mathbf{g}}t(\mathbf{h},\mathbf{k}))]^* U_{\mathbf{g}}(\,^{\mathbf{g}}t(\mathbf{h},\mathbf{kl}),\,^{\mathbf{gh}}t(\mathbf{k},\mathbf{l})) \\
& F^{t(\mathbf{ghk},\mathbf{l}),t(\mathbf{gh},\mathbf{k}),t(\mathbf{g},\mathbf{h})}[F^{t(\mathbf{ghk},\mathbf{l}),t(\mathbf{g},\mathbf{hk}),\,^{\mathbf{g}}t(\mathbf{h},\mathbf{k})}]^* \\
& F^{t(\mathbf{g},\mathbf{hkl}),\,^{\mathbf{g}}t(\mathbf{hk},\mathbf{l}),\,^{\mathbf{g}}t(\mathbf{h},\mathbf{k})}[F^{t(\mathbf{g},\mathbf{hkl}),\,^{\mathbf{g}}t(\mathbf{h},\mathbf{kl}),\,^{\mathbf{gh}}t(\mathbf{k},\mathbf{l})}]^* \\
& F^{t(\mathbf{gh},\mathbf{kl}),t(\mathbf{g},\mathbf{h}),\,^{\mathbf{gh}}t(\mathbf{k},\mathbf{l})}[F^{t(\mathbf{gh},\mathbf{kl}),\,^{\mathbf{gh}}t(\mathbf{k},\mathbf{l}),t(\mathbf{g},\mathbf{h})}]^*.
\end{aligned} \tag{44}$$

In Eq. (43) we have for ease of notation dropped the explicit indices for the $\tilde{F}$ symbols. Note that in Eq. (44), we have dropped labels in the data whenever they are fixed by fusion outcomes. For example, $U_{\mathbf{g}}(a, b; c)$ is written as $U_{\mathbf{g}}(a, b)$ if $c$ is uniquely determined by $a$ and $b$. Similarly $F^{abc}_{def}$ is written as $F^{abc}$ when $a,b,c$ are Abelian, as $d,e,f$ are then fixed uniquely by the fusion rules.

Eq. (43) shows that while the original theory was consistent, the new theory, whose defect $F$-symbols are given by $\tilde{F}$, may not satisfy its pentagon equation, up to an obstruction defined by $\mathfrak{O}_r(\mathbf{g},\mathbf{h},\mathbf{k},\mathbf{l})$. The new theory is therefore anomalous, while the original theory was consistent.

On the other hand, suppose that the original theory is not fully consistent. In this case, following the 15 moves of Fig. 8 does not give an equality, but rather an equality up to a 4-cochain $\mathcal{O}(\mathbf{g},\mathbf{h},\mathbf{k},\mathbf{l})$:

$$\tilde{F}\tilde{F}\mathcal{O}(\mathbf{g},\mathbf{h},\mathbf{k},\mathbf{l})\mathfrak{O}_r(\mathbf{g},\mathbf{h},\mathbf{k},\mathbf{l}) = \sum \tilde{F}\tilde{F}\tilde{F}. \tag{45}$$

Here $\mathcal{O}(\mathbf{g},\mathbf{h},\mathbf{k},\mathbf{l})$ detects the failure of the original theory from satisfying the consistency equation required by Fig. 8. We see then that $\mathfrak{O}_r(\mathbf{g},\mathbf{h},\mathbf{k},\mathbf{l})$ describes a *relative* anomaly, as it describes the failure of the new theory from satisfying its pentagon equation, relative to an anomaly $\mathcal{O}(\mathbf{g},\mathbf{h},\mathbf{k},\mathbf{l})$ of the original theory.

## 4.1 $O_r(\mathbf{g},\mathbf{h},\mathbf{k},\mathbf{l})$, $\mathcal{H}^4[G, \mathrm{U}(1)]$, and SPTs

Two comments are now in order. First, we expect that $\mathfrak{O}_r(\mathbf{g},\mathbf{h},\mathbf{k},\mathbf{l})$ is a 4-cocycle; we expect this should be provable using only the $G$-crossed consistency equations described in Ref. [14], however we do not pursue this further here.

Second, we assert that $\mathfrak{O}_r(\mathbf{g},\mathbf{h},\mathbf{k},\mathbf{l})$ can always be canceled completely by considering the theory to exist at the (2+1)D surface of a (3+1)D invertible state. The cohomology class of $\mathfrak{O}_r$ in $\mathcal{H}^4[G, \mathrm{U}(1)]$ determines which SPT state is required in the bulk, relative to the SPT that was required to cancel the anomaly of the original theory. The proof of this would require developing a theory of the full (3+1)D system, which we leave for future work.

Figure 8: Consistency condition for derivation of relative anomaly. Anyon and defect lines are assumed to have arrows directed upwards.

# 5 Incorporating space-time reflection symmetries

Here we wish to extend the discussion above to the case where the symmetry group $G$ may contain space-time reflection symmetries. We will propose a simple modification of the ap-

proach in Sec. 3-4 for unitary space-time orientation-preserving symmetries. Our proposed modification leads to a similar formula for the relative anomaly (see Eq. (51)).

A defect associated with reflection symmetry corresponds to inserting a crosscap in the system, as shown in Fig. 9. By considering a flattened crosscap, we can consider this as a **g** defect branch line as well, on the same footing as **g** defects for unitary symmetries, with the exception that when an anyon crosses the defect line, it gets reflected in space in addition to being permuted to its counterpart.

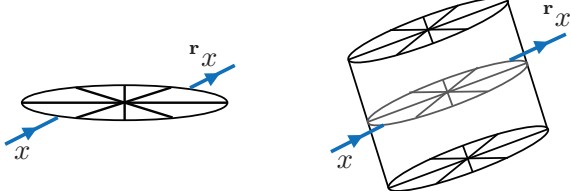

Figure 9: Left: crosscap, interpreted as a reflection symmetry defect, in space. We can consider flattening it and treating it on the same footing as other **g**-defects for unitary orientation-preserving symmetries. Right: in space-time, the reflection symmetry defect corresponds to a crosscap tube.

For time-reversing symmetries, which are anti-unitary, it is not clear whether one can define a sensible notion of a symmetry defect within a Hamiltonian formalism. However if we Wick rotate to imaginary time, then in Euclidean space-time we can treat time-reversal and spatial reflection symmetries on equal footing. Our usual notion of time-reversal corresponds in imaginary time to charge conjugation followed by reflection symmetry. Therefore, in this section we simply focus on spatial reflection symmetries. In subsequent sections, when discussing time-reversal symmetries, we then replace our formulas involving spatial reflection with the product of topological charge-conjugation and time-reversal.

## 5.1 Symmetry fractionalization classification

The symmetry fractionalization classification $\mathcal{H}^2_{[\rho]}(G, \mathcal{A})$ can now be rederived by considering the fusion of reflection symmetry defects, with the following important modification. When the anyon lines associated with **t** cross a reflection symmetry defect sheet, the line itself is reflected in space. This means that $\mathbf{t}(\mathbf{h}, \mathbf{k})$ transforms to $^{\mathbf{g}}\overline{\mathbf{t}(\mathbf{h}, \mathbf{k})}$ if it crosses a **g**-defect where **g** is spatial parity reversing. See for example Fig. 10. This implies the following non-trivial modification to the cocycle equation:

$$\mathbf{t}(\mathbf{g}, \mathbf{h})\mathbf{t}(\mathbf{gh}, \mathbf{k}) = {}^{\mathbf{g}}[\mathbf{t}(\mathbf{h}, \mathbf{k})^{p(\mathbf{g})}]\mathbf{t}(\mathbf{g}, \mathbf{hk}), \tag{46}$$

where $a^{p(\mathbf{g})} \equiv \bar{a}$ if **g** is spatial parity reversing. Remarkably, this same formula involving the charge conjugation operation associated with $p(\mathbf{g})$ was derived in Ref. [14] through completely different considerations, without introducing the notion of a reflection symmetry defect. We thus view the derivation of Eq. (46) from fusion of reflection symmetry defects as a non-trivial check on the validity of incorporating reflection symmetry defects into the algebraic description of defects.

## 5.2 Keeping track of local space-time orientation

To date, a complete consistent algebraic theory of fusion and braiding of symmetry defects involving reflection symmetry defects has not yet been developed. Here we propose an important

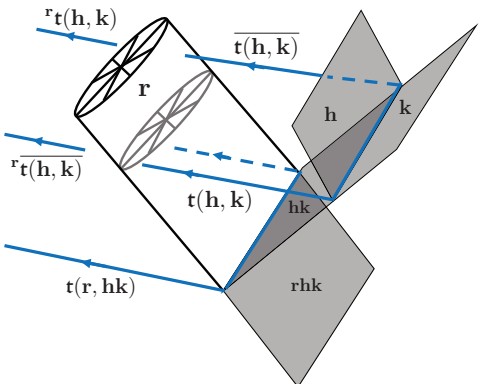

Figure 10: Fusion of three defect sheets, **r**, **h**, **k**, where **r** is a spatial reflection symmetry. As the anyon lines cross the crosscap sheet associated with **r**, they get reflected in space in addition to being permuted by the action of $\rho_{\mathbf{r}}$.

modification to the original $G$-crossed braided tensor category theory in order to incorporate reflection symmetry defects.

Our proposed modification is to first keep track of the local orientation of space-time in all regions. This can be done by labeling each region in between the defect lines by a local orientation $s = \pm$. For two regions separated by a defect line, if the defect is orientation-preserving (reversing), the orientations in the two regions are the same (opposite). Therefore, once the orientation in one region is known, orientations in all other regions are determined by the group elements associated with each of the defect lines. Note that a global space-time orientation is not in general well-defined in the presence of reflection defects; however given a local portion of a defect diagram, we can label the local space-time orientations.

Now, the $F$, $R$, $U$, and $\eta$ symbols all explicitly depend on the local orientations. We thus have $F_{def}^{abc}(\{s_i\})$, $R_c^{ab}(\{s_i\})$, $U_{\mathbf{g}}(a, b; c; \{s_i\})$, $\eta_a(\mathbf{g}, \mathbf{h}; \{s_i\})$, as shown in Fig. 11. Note that since the group labels on the defect lines specify the difference in local orientations across the line, it is sufficient to only specify the orientation in a single region.

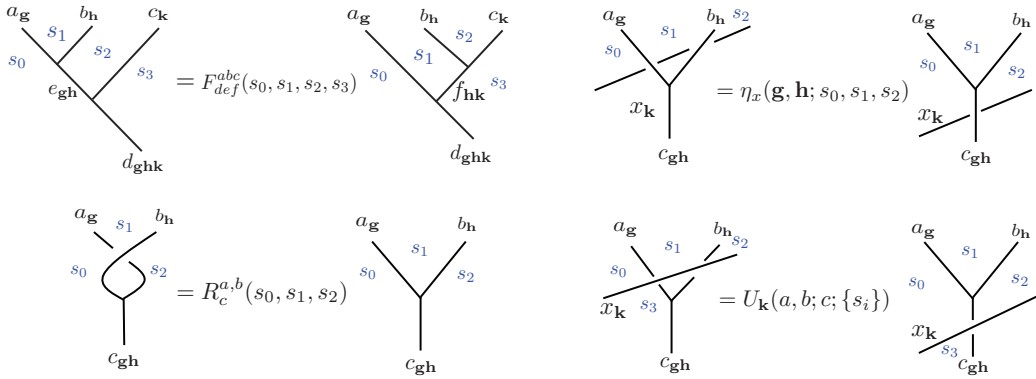

Figure 11: Including the local space-time orientations $\{s_i\}$ in the regions between the defect / anyon lines. This naturally gives the structure of a higher category. The lines are assumed to be directed with arrows pointing upwards.

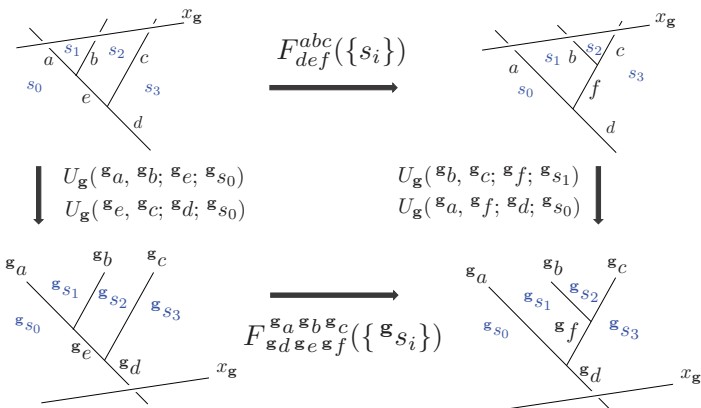

Figure 12: **g** action on the *F*-symbol diagram. All lines are assumed to be directed with arrows pointing upwards.

Furthermore we can consider a global symmetry action, $\rho_{\mathbf{g}}$ for $\mathbf{g} \in G$, on the whole diagram. Note this is the extension of $\rho_{\mathbf{g}}$ defined in Sec. 2.3, which acted only on the anyon theory, to an action on all of the defect data as well. Diagrammatically, this corresponds to sweeping a defect line associated with **g** across the diagram, as shown in Fig. 12 for the case of the *F*-symbol. We therefore have, for example,

$$
\begin{aligned}
\rho_{\mathbf{g}}[F^{abc}_{def}(\{s_i\})] = {} & U_{\mathbf{g}}(^{\mathbf{g}}a, {}^{\mathbf{g}}b; {}^{\mathbf{g}}e; {}^{\mathbf{g}}s_0)U_{\mathbf{g}}(^{\mathbf{g}}e, {}^{\mathbf{g}}c; {}^{\mathbf{g}}d; {}^{\mathbf{g}}s_0) \\
& F^{^{\mathbf{g}}a\,^{\mathbf{g}}b\,^{\mathbf{g}}c}_{^{\mathbf{g}}d\,^{\mathbf{g}}e\,^{\mathbf{g}}f}(^{\mathbf{g}}s_0) \\
& U_{\mathbf{g}}^{-1}(^{\mathbf{g}}b, {}^{\mathbf{g}}c; {}^{\mathbf{g}}f; {}^{\mathbf{g}}s_1)U_{\mathbf{g}}^{-1}(^{\mathbf{g}}a, {}^{\mathbf{g}}f; {}^{\mathbf{g}}d; {}^{\mathbf{g}}s_0),
\end{aligned}
\tag{47}
$$

where for ease of notation we have only included the top-left-most space-time index on the *U* symbols. The symmetry action on the local orientations is such that $^{\mathbf{g}}s = (-1)^{\sigma(\mathbf{g})}s$, where here $\sigma(\mathbf{g}) = 0$ if **g** is space-time parity even and $\sigma(\mathbf{g}) = 1$ if **g** is space-time parity odd.

In order for the theory to be symmetric, we wish that this *G* action keep invariant all of the data of the theory. Therefore, we impose the condition:

$$
\rho_{\mathbf{g}}[X(\cdots; s)] = X(\cdots; s),
\tag{48}
$$

where $X(\cdots; s)$ refers to any datum of the theory, such as the *F*, *R*, *U*, or $\eta$ symbols. This is equivalent to the condition that the corresponding diagram, for example Fig. 12, commutes, which ensures that one obtains the same results regardless of whether a defect line is swept across the diagram before or after the corresponding move.

Finally, in addition to Eq. (47), (48), we further impose that

$$
X(\cdots; {}^{\mathbf{g}}s) = K^{\sigma(\mathbf{g})}X(\cdots; s)K^{\sigma(\mathbf{g})},
\tag{49}
$$

where recall *K* refers to complex conjugation. One way to understand the appearance of this complex conjugation is as follows. Within a path integral formalism, a state on a space *M* corresponds to the path integral evaluated on a space-time *W* such that $\partial W = M$. Reversing the orientation of *M* corresponds to converting a state from a bra to a ket and vice versa. This requires a Hermitian conjugation to relate processes that occur before and after the reflection.

When space-time reflection symmetries are allowed, the pentagon equation must thus be modified, because we must keep track of the local orientations. The pentagon equation becomes:

$$
F^{fcd}_{egl}(s_0,s_2,s_3,s_4)F^{abl}_{efk}(s_0,s_1,s_2,s_4) = \sum_h F^{abc}_{gfh}(s_0,s_1,s_2,s_3)F^{ahd}_{egk}(s_0,s_1,s_3,s_4)F^{bcd}_{khl}(s_1,s_2,s_3,s_4).
\tag{50}
$$

Other consistency equations are similarly modified, however we will not explicitly consider them in this paper.

If we focus just on the fusion properties, which involve only the fusion rules and $F$ symbols, then the above structure no longer corresponds to a fusion category, because a fusion category does not require the additional data $\{s_i\}$. Rather, as described in a slightly different context in Ref. [21], such a fusion structure corresponds to the structure of a 2-category. The objects of the 2-category are the local orientations $\pm$, while the defects $a_{\mathbf{g}}$ are now considered to be 1-morphisms between the objects. The fusion and splitting processes that map $a_{\mathbf{g}} \times b_{\mathbf{h}} \to c_{\mathbf{gh}}$ then correspond to the 2-morphisms. Furthermore, the 2-category has a $G$-action, as described by the action of $\rho_{\mathbf{g}}$, and is also $G$-equivariant, because we require that the $G$ action leave the fusion rules and $F$ symbols invariant.

When we include the $G$-crossed braiding processes in addition to fusion, then the proper mathematical structure must be a $G$-equivariant 3-category with $G$ action. We leave a proper study of the relation between symmetry-enriched topological states and higher category theory for future work.

So far we have derived the action of $\rho_{\mathbf{g}}$ on the $F$-symbols and derived a modified pentagon equation. It is possible to derive the action of $\rho_{\mathbf{g}}$ on all of the data $\{F, R, U, \eta\}$, consistency conditions, and gauge transformations of the theory, which, combined with Eq. (48) and (49) thus provides a generalization of the $G$-crossed braided tensor category equations of Ref. [14] to space-time reflecting symmetries. We leave it for future work to fully derive all of these equations and to demonstrate their consistency and applicability to characterizing space-time reflection symmetric SETs.

Now, turning to the relative anomaly, precisely the same derivation as in Sec. 4 can be carried through, with the difference that now all of the data also includes a dependence on the local space-time orientations. Importantly this means that whenever an Abelian anyon associated with $\mathbf{t}$ crosses a $\mathbf{g}$ defect line, it gets conjugated to $^{\mathbf{g}}[\mathbf{t}^{p(\mathbf{g})}]$, as was the case for Eq. 46. We thus arrive at essentially the same formula as in the case without space-time reflections, with some minor modification:

$$
\begin{aligned}
\mathfrak{O}_r(\mathbf{g}, \mathbf{h}, \mathbf{k}, \mathbf{l}; s_0) = & R^{\mathbf{gh}[\mathbf{t}(\mathbf{k},\mathbf{l})^{p(\mathbf{gh})}], \mathbf{t}(\mathbf{g},\mathbf{h})}(s_0)\, \eta_{\mathbf{gh}[\mathbf{t}(\mathbf{k},\mathbf{l})^{p(\mathbf{gh})}]}(\mathbf{g}, \mathbf{h}, s_0) \\
& [U_{\mathbf{g}}(^{\mathbf{g}}[\mathbf{t}(\mathbf{hk},\mathbf{l})^{p(\mathbf{g})}], {}^{\mathbf{g}}[\mathbf{t}(\mathbf{h},\mathbf{k})^{p(\mathbf{g})}]; s_0)]^* U_{\mathbf{g}}(^{\mathbf{g}}[\mathbf{t}(\mathbf{h},\mathbf{kl})^{p(\mathbf{g})}], {}^{\mathbf{gh}}[\mathbf{t}(\mathbf{k},\mathbf{l})^{p(\mathbf{gh})}]; s_0) \\
& [F^{\mathbf{t}(\mathbf{ghk},\mathbf{l}), \mathbf{t}(\mathbf{g},\mathbf{hk}), {}^{\mathbf{g}}[\mathbf{t}(\mathbf{h},\mathbf{k})^{p(\mathbf{g})}]}(s_0)]^* F^{\mathbf{t}(\mathbf{g},\mathbf{hkl}), {}^{\mathbf{g}}[\mathbf{t}(\mathbf{hk},\mathbf{l})^{p(\mathbf{g})}], {}^{\mathbf{g}}[\mathbf{t}(\mathbf{h},\mathbf{k})^{p(\mathbf{g})}]}(s_0) \\
& [F^{\mathbf{t}(\mathbf{g},\mathbf{hkl}), {}^{\mathbf{g}}[\mathbf{t}(\mathbf{h},\mathbf{kl})^{p(\mathbf{g})}], {}^{\mathbf{gh}}[\mathbf{t}(\mathbf{k},\mathbf{l})^{p(\mathbf{gh})}]}(s_0)]^* F^{\mathbf{t}(\mathbf{gh},\mathbf{kl}), \mathbf{t}(\mathbf{g},\mathbf{h}), {}^{\mathbf{gh}}[\mathbf{t}(\mathbf{k},\mathbf{l})^{p(\mathbf{gh})}]}(s_0) \\
& [F^{\mathbf{t}(\mathbf{gh},\mathbf{kl}), {}^{\mathbf{gh}}[\mathbf{t}(\mathbf{k},\mathbf{l})^{p(\mathbf{gh})}], \mathbf{t}(\mathbf{g},\mathbf{h})}(s_0)]^* F^{\mathbf{t}(\mathbf{ghk},\mathbf{l}), \mathbf{t}(\mathbf{gh},\mathbf{k}), \mathbf{t}(\mathbf{g},\mathbf{h})}(s_0).
\end{aligned}
\tag{51}
$$

Note that, as mentioned above, since the defect group labels fix all the local orientations $s_i$ once $s_0$ is fixed, we only need to keep track of one additional variable, $s_0$, in the above formula, which corresponds to the local orientation in the top left of the associated diagram.

From Eqs. (48), (49), we see that $\mathfrak{O}_r(\mathbf{g}, \mathbf{h}, \mathbf{k}, \mathbf{l}; -)$ is fixed by $\mathfrak{O}_r(\mathbf{g}, \mathbf{h}, \mathbf{k}, \mathbf{l}; +)$. We thus define the relative anomaly

$$
\mathfrak{O}_r(\mathbf{g}, \mathbf{h}, \mathbf{k}, \mathbf{l}) \equiv \mathfrak{O}_r(\mathbf{g}, \mathbf{h}, \mathbf{k}, \mathbf{l}; +).
\tag{52}
$$

# 6 Time-reversal symmetry, $G = \mathbb{Z}_2^{\mathbf{T}}$

Here we apply the relative anomaly formula to the case where $G = \mathbb{Z}_2^{\mathbf{T}}$. Note that (3+1)D SPTs with $\mathbb{Z}_2^{\mathbf{T}}$ symmetry have a $\mathbb{Z}_2 \times \mathbb{Z}_2$ classification, while $\mathcal{H}^4[\mathbb{Z}_2^{\mathbf{T}}, \mathrm{U}(1)] = \mathbb{Z}_2$. Therefore the approach here only captures the relative anomaly that is within the group cohomology classification.

To do computations, we work in a "canonical gauge," where $\mathfrak{O}_r(\mathbf{g}_1, \mathbf{g}_2, \mathbf{g}_3, \mathbf{g}_4) = 1$ if any $\mathbf{g}_i = \mathbf{1}$, where $\mathbf{1}$ refers to the identity element of $G$. Any residual gauge transformation (that is, any shift of $\mathfrak{O}_r$ by a coboundary) $d\epsilon(\mathbf{g}_1, \mathbf{g}_2, \mathbf{g}_3, \mathbf{g}_4)$ must then have the property that $\epsilon(\mathbf{g}_1, \mathbf{g}_2, \mathbf{g}_3) = 1$ if any $\mathbf{g}_i = \mathbf{1}$.

One can then check that

$$\mathcal{I} \equiv \mathfrak{O}_r(\mathbf{T}, \mathbf{T}, \mathbf{T}, \mathbf{T}) \tag{53}$$

is invariant under any residual gauge transformations in the canonical gauge.

Next, we note that it is always possible to pick a gauge for a representative 2-cocycle $\mathbf{t}$ such that

$$t(\mathbf{1}, \mathbf{1}) = t(\mathbf{T}, \mathbf{1}) = t(\mathbf{1}, \mathbf{T}) = 1. \tag{54}$$

Here we use 1 to denote the vacuum sector of the anyon theory. Furthermore, we can also always pick a gauge where $U_{\mathbf{T}}(a, 1) = U_{\mathbf{T}}(1, a) = 1$ [14]. This choice of gauge satisfies our choice of canonical gauge for $\mathfrak{O}_r(\mathbf{g}_1, \mathbf{g}_2, \mathbf{g}_3, \mathbf{g}_4)$ described above.

Applying the relative anomaly formula, we then find

$$\mathcal{I} = \eta_{t(\mathbf{T}, \mathbf{T})}(\mathbf{T}, \mathbf{T}) \theta_{t(\mathbf{T}, \mathbf{T})}. \tag{55}$$

Recall that when $^{\mathbf{T}}a = a$, $\eta_a^{\mathbf{T}} \equiv \eta_a(\mathbf{T}, \mathbf{T})$ is a gauge-invariant symmetry fractionalization quantum number that indicates whether $a$ carries a "local" Kramers degeneracy; that is, whether $\mathbf{T}^2 = -1$ "locally" for the anyon $a$ [14].

## 6.1 Relation to absolute anomaly indicator, $\mathcal{Z}(\mathbb{RP}^4)$

Recently, the absolute anomaly for $\mathbb{Z}_2^{\mathbf{T}}$ was computed in general by computing the path integral of the (3+1)D theory on $\mathbb{RP}^4$ [21]:

$$\mathcal{Z}(\mathbb{RP}^4) = \frac{1}{\mathcal{D}} \sum_{a|a = {}^{\mathbf{T}}a} \eta_a^{\mathbf{T}} \theta_a d_a. \tag{56}$$

This formula was independently conjectured as an "anomaly indicator" in Ref. [24]. Here we demonstrate compatibility of the relative anomaly, Eq. (55), with the absolute anomaly, Eq. (56)

As discussed in Sec. 3, shifting a symmetry fractionalization class by a 2-cocyle $[\mathbf{t}]$ induces a shift in the $\eta$ symbols: $\eta \to \eta'$, with

$$\eta_a'(\mathbf{T}, \mathbf{T}) = \eta_a(\mathbf{T}, \mathbf{T}) M_{a, t(\mathbf{T}, \mathbf{T})}. \tag{57}$$

That this also holds for anti-unitary time-reversal symmetry was demonstrated explicitly in Ref. [14].

Let us define

$$\mathbf{t} \equiv t(\mathbf{T}, \mathbf{T}), \tag{58}$$

as this is the only non-trivial element. The 2-cocycle condition requires

$$^{\mathbf{T}}\mathbf{t} = \mathbf{t}. \tag{59}$$

We have:

$$\eta_a' \theta_a \eta_\mathbf{t} \theta_\mathbf{t} = \eta_a M_{a\mathbf{t}} \theta_a \eta_\mathbf{t} \theta_\mathbf{t} = \eta_{a\mathbf{t}} \theta_{a\mathbf{t}}. \tag{60}$$

Letting $\mathcal{Z}'(\mathbb{RP}^4)$ be given by Eq. (56) with the $\eta'$ quantum numbers, we find

$$
\begin{aligned}
\mathcal{Z}'(\mathbb{RP}^4)\theta_\mathbf{t}\eta_\mathbf{t} &= \frac{1}{\mathcal{D}} \sum_{a|a={}^\mathbf{T}a} \eta'_a \theta_a d_a \theta_\mathbf{t} \eta_\mathbf{t} \\
&= \frac{1}{\mathcal{D}} \sum_{a|a={}^\mathbf{T}a} \eta_{a\mathbf{t}} \theta_{a\mathbf{t}} d_{a\mathbf{t}} \\
&= \frac{1}{\mathcal{D}} \sum_{a|(a\mathbf{t})={}^\mathbf{T}(a\mathbf{t})} \eta_{a\mathbf{t}} \theta_{a\mathbf{t}} d_{a\mathbf{t}} \\
&= \frac{1}{\mathcal{D}} \sum_{x|x={}^\mathbf{T}x} \eta_x \theta_x d_x \\
&= \mathcal{Z}(\mathbb{RP}^4).
\end{aligned}
\tag{61}
$$

Note we have used ${}^\mathbf{T}\mathbf{t} = \mathbf{t}$ by the cocycle condition. Therefore we have proven

$$
\mathcal{Z}'(\mathbb{RP}^4)\theta_\mathbf{t}\eta_\mathbf{t} = \mathcal{Z}(\mathbb{RP}^4),
\tag{62}
$$

which confirms the relative anomaly formula.

We note that for Abelian topological states, $\mathbb{Z}_2^\mathbf{T}$ anomalies were further studied in Ref. [34], where it was shown that the absolute anomaly formula Eq. (56) reduces to the Arf invariant of a certain quadratic form $q$ defined as follows. Define the Abelian group $C = \mathrm{Ker}(1-\mathbf{T})/\mathrm{Im}(1+\mathbf{T})$, which is the group of all $\mathbf{T}$ invariant anyons, $a = {}^\mathbf{T}a$, modulo those that are of the form $a = c \times {}^\mathbf{T}c$, for some anyon $c$. Then one defines $q(a) = \theta_a \eta_a^\mathbf{T}$, considered as a function on $C$. The relative anomaly formula derived in this paper, in the case of Abelian topological phases, is a well-known property of Arf invariants.

# 7  $\mathrm{U}(1) \times \mathbb{Z}_2^\mathbf{T}$ and $\mathrm{U}(1) \rtimes \mathbb{Z}_2^\mathbf{T}$ symmetry

Here we analyze the cases where the symmetry group corresponds to $\mathrm{U}(1) \times \mathbb{Z}_2^\mathbf{T}$ or $\mathrm{U}(1) \rtimes \mathbb{Z}_2^\mathbf{T}$ symmetry. We denote elements of $\mathrm{U}(1) \times \mathbb{Z}_2^\mathbf{T}$ and $\mathrm{U}(1) \rtimes \mathbb{Z}_2^\mathbf{T}$ as $(U_\theta, \mathbf{g})$, where $\theta \in [0, 2\pi)$ and $\mathbf{g} \in \{1, \mathbf{T}\}$. The multiplication is

$$
(U_\alpha, \mathbf{g}) \cdot (U_\beta, \mathbf{h}) = (U_{\alpha+\beta}, \mathbf{gh})
\tag{63}
$$

for $\mathrm{U}(1) \times \mathbb{Z}_2^\mathbf{T}$ and

$$
(U_\alpha, \mathbf{g}) \cdot (U_\beta, \mathbf{h}) = (U_{\alpha+q(\mathbf{g})\beta}, \mathbf{gh})
\tag{64}
$$

for $\mathrm{U}(1) \rtimes \mathbb{Z}_2^\mathbf{T}$. Below, for convenience we will also sometimes denote $(U_\alpha, \mathbf{g})$ as $U_\alpha \mathbf{g}$, for $\mathbf{g} \in \{1, \mathbf{T}\}$.

The anomaly classification, given by bulk (3+1)D SPTs, is given by:

$$
\begin{aligned}
\mathcal{H}^4[\mathrm{U}(1) \times \mathbb{Z}_2^\mathbf{T}, \mathrm{U}(1)] &= \mathbb{Z}_2^3, \\
\mathcal{H}^4[\mathrm{U}(1) \rtimes \mathbb{Z}_2^\mathbf{T}, \mathrm{U}(1)] &= \mathbb{Z}_2^2.
\end{aligned}
\tag{65}
$$

There is also an additional $\mathbb{Z}_2$ coming from a "beyond cohomology" SPT in this case, associated with time-reversal symmetry alone, whose anomaly can be detected by $e^{2\pi i c_-/8} = \pm 1$, where $c_-$ is the chiral central charge. Furthermore, one of the $\mathbb{Z}_2$ factors in the group cohomology classifications above is associated with pure time-reversal symmetry and corresponds to the case studied in Sec. 6. Therefore we see the appearance of two additional independent anomalies for $\mathrm{U}(1) \times \mathbb{Z}_2^\mathbf{T}$ symmetry and one for $\mathrm{U}(1) \rtimes \mathbb{Z}_2^\mathbf{T}$ symmetry. Note that these are mixed anomalies, because $\mathrm{U}(1)$ by itself has trivial fourth cohomology.

In fact it is already known that one of the mixed anomalies for both $U(1) \times \mathbb{Z}_2^{\mathbf{T}}$ and $U(1) \rtimes \mathbb{Z}_2^{\mathbf{T}}$ symmetry corresponds to whether the vison, which corresponds to the excitation obtained by threading $2\pi$ units of U(1) flux, is a fermion. For $U(1) \times \mathbb{Z}_2^{\mathbf{T}}$, the second mixed anomaly corresponds to whether the vison is a Kramers singlet or doublet. These diagnostics were used recently in Ref. [27] to derive formulas for absolute anomaly indicators for $U(1) \times \mathbb{Z}_2^{\mathbf{T}}$ and $U(1) \rtimes \mathbb{Z}_2^{\mathbf{T}}$ symmetry groups. Below we apply our relative anomaly formula to compare with these results.

## 7.1 Symmetry fractionalization

First we explain physically the fractionalization classes of $U(1) \times \mathbb{Z}_2^{\mathbf{T}}$. One can show that there are basically two pieces of information: first of all, an anyon $a$ can carry a fractional charge $q_a$ (defined mod 1) under U(1). This is captured by a (absolute) vison $v_0 \in \mathcal{A}$ such that

$$e^{2\pi i q_a} = M_{v_0, a}. \tag{66}$$

Next, we must specify the action of the time-reversal symmetry $\mathbf{T}$, including the permutation of anyon types $\rho_{\mathbf{T}}$ and the local $\mathbf{T}^2$ value $\eta_a^{\mathbf{T}}$ for all $\mathbf{T}$-invariant anyons.

The 2-cocycle $[\mathbf{t}] \in \mathcal{H}_{[\rho]}^2(U(1) \times \mathbb{Z}_2^{\mathbf{T}}, \mathcal{A})$ can be generally parametrized as

$$\mathbf{t}\big((U_\alpha, \mathbf{g}), (U_\beta, \mathbf{h})\big) = \mathbf{t}(\mathbf{g}, \mathbf{h}) v^{([\alpha]_{2\pi} + [\beta]_{2\pi} - [\alpha+\beta]_{2\pi})/2\pi}. \tag{67}$$

where $[\alpha]_{2\pi}$ denotes $\alpha$ modulo $2\pi$. Here $\mathbf{t}(\mathbf{g}, \mathbf{h})$ is a 2-cocycle associated with $\mathcal{H}_{[\rho]}^2(\mathbb{Z}_2^{\mathbf{T}}, \mathcal{A})$, while $v^{([\alpha]_{2\pi} + [\beta]_{2\pi} - [\alpha+\beta]_{2\pi})/2\pi}$ gives a 2-cocycle associated with $\mathcal{H}^2(U(1), \mathcal{A})$. This form for the 2-cocycle follows because the Künneth formula in this case gives $\mathcal{H}_{[\rho]}^2(U(1) \times \mathbb{Z}_2^{\mathbf{T}}, \mathcal{A}) = \mathcal{H}_{[\rho]}^2(\mathbb{Z}_2^{\mathbf{T}}, \mathcal{A}) \oplus \mathcal{H}^2(U(1), \mathcal{A})$.

The 2-cocycle condition is satisfied if both $\mathbf{t}(\mathbf{T}, \mathbf{T})$ and $v$ are invariant under $\mathbf{T}$. $v$ here has the interpretation of the anyon resulting from a $2\pi$ flux insertion in the new theory, relative to the old theory. This can be seen by looking at $\mathbf{g} = \mathbf{h} = 1, \beta = 2\pi - \alpha$, which gives

$$\mathbf{t}\big((U_\alpha, \mathbf{1}), (U_{2\pi-\alpha}, \mathbf{1})\big) = v. \tag{68}$$

In other words, we may refer to $v$ here as a "relative vison."

Because $\mathbf{T}$ commutes with U(1), the $2\pi$ flux and thus $v$ is invariant under $\mathbf{T}$:

$$^{\mathbf{T}}v = v. \tag{69}$$

On the other hand, for $U(1) \rtimes \mathbb{Z}_2^{\mathbf{T}}$, one can show that the 2-cocycle $[\mathbf{t}] \in \mathcal{H}_{[\rho]}^2(U(1) \rtimes \mathbb{Z}_2^{\mathbf{T}}, \mathcal{A})$ can be generally parametrized as

$$\mathbf{t}\big((U_\alpha, \mathbf{g}), (U_\beta, \mathbf{h})\big) = \mathbf{t}(\mathbf{g}, \mathbf{h}) v^{([\alpha]_{2\pi} + s(\mathbf{g})[\beta]_{2\pi} - [\alpha+s(\mathbf{g})\beta]_{2\pi})/2\pi}, \tag{70}$$

where $s(\mathbf{g}) = 1 - 2q(\mathbf{g}) = \pm 1$ depending on whether $\mathbf{g}$ is time-reversing. Here the relative vison $v$ must be a self-dual Abelian anyon and invariant under $\mathbf{T}$, as a $2\pi$ flux turns into a $-2\pi$ flux under $\mathbf{T}$.

## 7.2 Cohomology invariants

We first describe a set of invariants for the $\mathcal{H}^4$ cohomology classes. Let us first focus on $\mathcal{H}^4[U(1) \times \mathbb{Z}_2^{\mathbf{T}}, U(1)] = \mathbb{Z}_2^3$. One quick way to understand the invariants is to consider the $\mathbb{Z}_2 \times \mathbb{Z}_2^{\mathbf{T}}$ subgroup, where the $\mathbb{Z}_2$ is generated by $U_\pi$. Notice that besides $\mathbf{T}$ there is another

order-2 anti-unitary element $\mathbf{T}' = U_\pi \mathbf{T}$. Thus we can define a pure time-reversal anomaly for both $\mathbf{T}$ and $\mathbf{T}'$:

$$
\begin{aligned}
\mathcal{I}_1 &= \mathfrak{O}_r(\mathbf{T}, \mathbf{T}, \mathbf{T}, \mathbf{T}), \\
\mathcal{I}_2 &= \mathfrak{O}_r(\mathbf{T}', \mathbf{T}', \mathbf{T}', \mathbf{T}').
\end{aligned}
\tag{71}
$$

For the third $\mathbb{Z}_2$ invariant, we consider the following expression:

$$
\mathcal{I}_3 = \frac{\mathfrak{O}_r(\mathbf{T}, U_\pi, U_\pi, U_\pi) \mathfrak{O}_r(U_\pi, U_\pi, \mathbf{T}, U_\pi)}{\mathfrak{O}_r(U_\pi, \mathbf{T}, U_\pi, U_\pi) \mathfrak{O}_r(U_\pi, U_\pi, U_\pi, \mathbf{T})} \mathfrak{O}_r(U_\pi, U_\pi, U_\pi, U_\pi).
\tag{72}
$$

The form of the expression is motivated by relation of the slant product of the 4-cocycle to a time-reversal (i.e. $\mathbf{T}$) domain wall. The bulk bosonic topological insulator can be viewed as proliferation of time-reversal domain walls decorated with U(1) bosonic integer quantum Hall (BIQH) states with Hall conductance $\sigma_{xy} = 2$. If the U(1) symmetry is broken down to $\mathbb{Z}_2$, then the BIQH state becomes the $\mathbb{Z}_2$ Levin-Gu SPT state [35]. The invariant $\mathcal{I}_3$ is designed to detect such decorated 2D SPT states on time-reversal domain walls.

Note that these expressions are all assuming a canonical gauge for $\mathfrak{O}_r$, as described in the previous section, and thus are invariant under residual gauge transformations that preserve this canonical gauge.

For $G = \mathrm{U}(1) \rtimes \mathbb{Z}_2^\mathbf{T}$, the above expressions are all still invariants, however we will see that $\mathcal{I}_2 = \mathcal{I}_1$. Therefore $\mathcal{I}_1, \mathcal{I}_3$ are the two invariants that distinguish the $\mathbb{Z}_2^2$ cohomology classes.

### 7.3 General anomaly formula

Define $t = \mathfrak{t}(\mathbf{T}, \mathbf{T})$. A straightforward application of Eq. (71), (72) and the relative anomaly formula yields the following results for $G = \mathrm{U}(1) \times \mathbb{Z}_2^\mathbf{T}$:

$$
\begin{aligned}
\mathcal{I}_1 &= \theta_t \eta_t(\mathbf{T}, \mathbf{T}), \\
\mathcal{I}_2 &= \theta_{tv} \eta_{tv}(\mathbf{T}', \mathbf{T}'), \\
\mathcal{I}_3 &= \theta_v \frac{\eta_v(\mathbf{T}, U_\pi)}{\eta_v(U_\pi, \mathbf{T})} \eta_v(U_\pi, U_\pi).
\end{aligned}
\tag{73}
$$

Since $v$ is invariant under $\mathbf{T}$, we have the following two cocycle conditions:

$$
\begin{aligned}
\eta_v(U_\pi, \mathbf{T})\eta_v(U_\pi \mathbf{T}, U_\pi \mathbf{T}) &= \eta_v(U_\pi, U_\pi)\eta_v(\mathbf{T}, U_\pi \mathbf{T}), \\
\eta_v(\mathbf{T}, \mathbf{T}) &= \eta_v(\mathbf{T}, U_\pi \mathbf{T})\eta_v^{-1}(\mathbf{T}, U_\pi).
\end{aligned}
\tag{74}
$$

Thus we find

$$
\frac{\eta_v(\mathbf{T}, U_\pi)}{\eta_v(U_\pi, \mathbf{T})} \eta_v(U_\pi, U_\pi) = \frac{\eta_v(\mathbf{T}', \mathbf{T}')}{\eta_v(\mathbf{T}, \mathbf{T})}.
\tag{75}
$$

We can also show that $\eta_{tv}(\mathbf{T}', \mathbf{T}') = \eta_t(\mathbf{T}', \mathbf{T}')\eta_v(\mathbf{T}', \mathbf{T}')$, which follows from the fact that both $t$ and $v$ should be invariant under $\mathbf{T}$ (and thus $\mathbf{T}'$).

To compare the results with more familiar characterization of the anomalies, we need to determine the absolute anomaly. In this case, we can choose a reference SET state in the following way:

1. Since U(1) does not permute anyons, there is a canonical choice for a reference where U(1) acts completely trivially, namely all anyons are charge-neutral, so that the vison is the identity. This implies that there is no mixed anomaly involving the U(1) symmetry.

2. As shown in Ref. [14], the time-reversal symmetry fractionalization can be characterized by how $\mathbf{T}$ permutes anyons and the local $\mathbf{T}^2$ values of $\mathbf{T}$-invariant anyons, $\eta_a(\mathbf{T}, \mathbf{T})$ for $^{\mathbf{T}}a = a$. It is believed that this gives a complete characterization of time-reversal symmetry fractionalization, although so far this has not been proven. We assume that there is no symmetry localization anomaly (characterized by a non-trivial element in $\mathcal{H}^3_{[\rho]}(G, \mathcal{A})$ for the $\mathbf{T}$ action [14, 32]). We choose a reference state which has no $\mathbb{Z}^{\mathbf{T}}_2$ anomaly.

3. For the symmetry groups $U(1) \times \mathbb{Z}^{\mathbf{T}}_2$ or $U(1) \rtimes \mathbb{Z}^{\mathbf{T}}_2$, the permutation action for $\mathbf{T}'$ must be identical to that of $\mathbf{T}$. Furthermore, because $U(1)$ rotations, including $U_\pi$, act trivially on anyons, we have $\eta_a(\mathbf{T}', \mathbf{T}') = \eta_a(\mathbf{T}, \mathbf{T})$ in this reference state, which also implies that there is no anomaly associated with $\mathbf{T}'$ alone.

Now compared to this anomaly-free reference SET state, the invariants become

$$
\begin{aligned}
\mathcal{I}_1 &= \theta_t \eta^{\mathbf{T}}_t, \\
\mathcal{I}_2 &= \theta_t \eta^{\mathbf{T}}_t \theta_v M_{tv} \eta^{\mathbf{T}}_v, \\
\mathcal{I}_3 &= \theta_v.
\end{aligned}
\tag{76}
$$

It is more convenient to replace $\mathcal{I}_2$ with $\mathcal{I}'_2 = \frac{\mathcal{I}_2}{\mathcal{I}_1 \mathcal{I}_3}$:

$$
\begin{aligned}
\mathcal{I}_1 &= \theta_t \eta^{\mathbf{T}}_t, \\
\mathcal{I}'_2 &= M_{tv} \eta^{\mathbf{T}}_v, \\
\mathcal{I}_3 &= \theta_v.
\end{aligned}
\tag{77}
$$

Now $\mathcal{I}'_2$ has the interpretation of the local $\mathbf{T}^2$ value for $v$ in the new SET phase, which is the reference one modified by the fractionalization class $\mathbf{t}$. $\mathcal{I}_3$ is determined by whether the vison is a boson or fermion. These are precisely the known anomaly indicators for $U(1) \times \mathbb{Z}^{\mathbf{T}}_2$ symmetry [27].

Let us apply these formulas to an example, where we reproduce the results of [30]. Consider a $\mathbb{Z}_2$ toric code topological order, and suppose $\mathbf{T}$ does not permute anyons. In this case, the relative anomalies agree with the "absolute" ones, and $\eta_a(\mathbf{g}, \mathbf{h}) = 1$ in the reference SET.

$$
\mathcal{I}_1 = \theta_t, \quad \mathcal{I}_2 = M_{v,t}, \quad \mathcal{I}_3 = \theta_v.
\tag{78}
$$

For the $eCmC$ state, which is the state where both the $e$ and the $m$ particle carry half-charge under the $U(1)$ symmetry and are Kramers singlets, we set $t = 1, v = \psi$, and thus $(\mathcal{I}_1, \mathcal{I}_2, \mathcal{I}_3) = (1, 1, -1)$. For the $eCmT$ state, which is the state where the $e$ particle carries half-charge and is a Kramers singlet, while the $m$ particle carries integer charge and is a Kramers doublet, we set $\mathbf{t}(\mathbf{T}, \mathbf{T}) = e, v = m$, which yields $(\mathcal{I}_1, \mathcal{I}_2, \mathcal{I}_3) = (1, -1, 1)$. This also shows explicitly that two mixed anomalies associated with $\mathcal{I}_2$ and $\mathcal{I}_3$ are independent.

Let us now turn to the case with $U(1) \rtimes \mathbb{Z}^{\mathbf{T}}_2$ symmetry. In this case the group cohomology classification gives $\mathcal{H}^4[U(1) \rtimes \mathbb{Z}^{\mathbf{T}}_2, U(1)] = \mathbb{Z}^2_2$. There is only one mixed anomaly, which corresponds to the whether the vison is a fermion. We can still use the invariants for the $\mathbb{Z}_2 \times \mathbb{Z}^{\mathbf{T}}_2$ subgroup that we used in the $U(1) \times \mathbb{Z}^{\mathbf{T}}_2$ case, but now notice that $\mathbf{t}(U_\pi \mathbf{T}, U_\pi \mathbf{T}) = \mathbf{t}(\mathbf{T}, \mathbf{T})$, and therefore $\mathcal{I}_2 = \mathcal{I}_1$. Using the same convention for the reference SET phase, we obtain the following anomaly indicators:

$$
\mathcal{I}_1 = \theta_t \eta^{\mathbf{T}}_t, \mathcal{I}_3 = \theta_v,
\tag{79}
$$

which again agrees with known results [27].

# 8 $\mathbb{Z}_4^{\mathbf{T}}$ symmetry

In this section we study the case of $\mathbb{Z}_4^{\mathbf{T}}$, which has been out of reach using previous methods. In this case, $\mathbf{T}^2$ is a non-trivial unitary symmetry. Physically, $\mathbf{T}$ can correspond to some combination of the true time-reversal operation of a physical system together with a non-trivial unitary symmetry. For example, we can consider a bosonic system where $\mathbf{T}^2 = (-1)^{N_b}$, where $N_b$ is the boson number.

## 8.1 $\mathbb{Z}_4^{\mathbf{T}}$ SPTs in (1+1)D and $\mathcal{H}^2[\mathbb{Z}_4^{\mathbf{T}}, U(1)]$

In (2+1)D systems, $\mathbb{Z}_2^{\mathbf{T}}$ symmetry fractionalization can fruitfully be understood by dimensional reduction and considering (1+1)D $\mathbb{Z}_2^{\mathbf{T}}$ SPTs [21,36]. It is natural therefore to begin by studying (1+1)D $\mathbb{Z}_4^{\mathbf{T}}$ SPTs.

In (1+1)D, $\mathbb{Z}_4^{\mathbf{T}}$ SPTs have a $\mathbb{Z}_2$ classification, corresponding to the cohomology group

$$\mathcal{H}^2[\mathbb{Z}_4^{\mathbf{T}}, U(1)] = \mathbb{Z}_2. \tag{80}$$

$\mathcal{H}^2[\mathbb{Z}_4^{\mathbf{T}}, U(1)]$ characterizes projective representations of $\mathbb{Z}_4^{\mathbf{T}}$. Physically, this means that a non-trivial $\mathbb{Z}_4^{\mathbf{T}}$ SPT state on a 1D space with boundary has a symmetry-protected degenerate two-dimensional subspace at each edge, forming a projective representation of $\mathbb{Z}_4^{\mathbf{T}}$.

One possible manifestation of a physical system with $\mathbb{Z}_4^{\mathbf{T}}$ symmetry is a system of bosons where $\mathbf{T}^2 = (-1)^{N_b}$, and where $N_b$ is the boson number. In this case, Kramers theorem implies that each boson, which carries a linear representation of $\mathbb{Z}_4^{\mathbf{T}}$, must carry a local Kramers degeneracy. However, the projective representation of $\mathbb{Z}_4^{\mathbf{T}}$ is also two-dimensional. (A generator for it can be taken to be $V_{\mathbf{T}} = e^{-i\sigma^y \pi/4} K$). It is interesting in this case that the linear and projective representations have the same dimension, but nevertheless are fundamentally distinct from each other. The non-trivial projective representation of $\mathbb{Z}_4^{\mathbf{T}}$ can be thought of as carrying fractional $\mathbb{Z}_2$ charge under the unitary symmetry $\mathbf{T}^2$.

It is useful for future reference to note that the following combination of 2-cocyles $\omega_2(\mathbf{g}, \mathbf{h}) \in U(1)$ is invariant under gauge transformations (i.e. under shifting $\omega_2$ by a 2-coboundary):

$$\eta^{\mathbf{T}} = \frac{\omega_2(\mathbf{T}, \mathbf{T}^2)\omega_2(\mathbf{T}^2, \mathbf{T}^2)}{\omega_2(\mathbf{T}^2, \mathbf{T})}. \tag{81}$$

## 8.2 $\mathbb{Z}_4^{\mathbf{T}}$ symmetry fractionalization in (2+1)D

As reviewed in Sec. 2,3, symmetry fractionalization is characterized in terms of distinct, gauge-inequivalent consistent choices of the $\{\eta\}$ and $\{U\}$ symbols [14]. For $\mathbb{Z}_4^{\mathbf{T}}$ symmetry, we will discuss two types of gauge-invariant quantum numbers associated to certain anyons.

First, when $^{\mathbf{T}}a = a$, one can also define an invariant

$$\eta_a^{\mathbf{T}} \equiv \frac{\eta_a(\mathbf{T}, \mathbf{T}^2)\eta_a(\mathbf{T}^2, \mathbf{T}^2)}{\eta_a(\mathbf{T}^2, \mathbf{T})}. \tag{82}$$

Under a symmetry action gauge transformation (see Sec. 2.4), $\eta_a^{\mathbf{T}} \to \eta_a^{\mathbf{T}} \frac{1}{(\gamma_{\mathbf{T}a}(\mathbf{T}^2))^* \gamma_a(\mathbf{T}^2)}$. Thus $\eta_a^{\mathbf{T}}$ is gauge invariant when $^{\mathbf{T}}a = a$. Note that the definition is nothing but the invariant that detects a nontrivial $\mathbb{Z}_4^{\mathbf{T}}$ projective representation, defined in the previous section.

On the other hand, $\mathbf{T}^2$ generates a unitary $\mathbb{Z}_2$ subgroup, and anyons can carry fractional $\mathbb{Z}_2$ charges. A general definition of fractional $\mathbb{Z}_2$ charge will be given in Sec. 9.1. Here we consider a special case: self-dual anyons $a$ (i.e. the fusion of $a$ satisfies $a \times a = 1 + \cdots$), which

are also invariant under $\mathbf{T}^2$ (but not necessarily $\mathbf{T}$). We define an invariant $\lambda_a^{\mathbf{T}^2}$ measuring whether $a$ carries fractional $\mathbb{Z}_2$ charge under $\mathbf{T}^2$:

$$\lambda_a^{\mathbf{T}^2} \equiv \eta_a(\mathbf{T}^2, \mathbf{T}^2) U_{\mathbf{T}^2}(a, a; 1). \tag{83}$$

It is straightforward to verify that this expression is gauge invariant. This is a special case of Eq. (106).

We note that invariants for $\mathbb{Z}_2$ symmetry fractionalization can also be defined for anyons that satisfy $\bar{a} = {}^{\mathbf{T}^2}a$ (see Eq. (107)). This kind of $\mathbb{Z}_2$ charge will be used in the example discussed in Sec. 8.5.5.

We now prove that $\eta_a^{\mathbf{T}} = \lambda_a^{\mathbf{T}^2}$ for a self-dual, $\mathbf{T}$-invariant $a$. This amounts to the following identity:

$$\frac{\eta_a(\mathbf{T}, \mathbf{T}^2)}{\eta_a(\mathbf{T}^2, \mathbf{T})} = U_{\mathbf{T}^2}(a, a; 1). \tag{84}$$

First, consider the fusion channel $a \times a \to 1$. We have

$$\eta_a(\mathbf{T}, \mathbf{T})^2 = U_{\mathbf{T}^2}(a, a; 1). \tag{85}$$

Then consider the 2-cocycle condition with $\mathbf{T}, \mathbf{T}, \mathbf{T}$:

$$\eta_a(\mathbf{T}, \mathbf{T}) \eta_a(\mathbf{T}^2, \mathbf{T}) = \eta_a(\mathbf{T}, \mathbf{T}^2) \eta_a(\mathbf{T}, \mathbf{T})^{-1}. \tag{86}$$

Combining the two relations immediately gives the desired identity.

Now we study $[\mathbf{t}] \in \mathcal{H}_\rho^2[\mathbb{Z}_4, \mathcal{A}]$. We denote $\mathbb{Z}_4 = \{0, 1, 2, 3\}$. One can show that a general 2-cocycle can always be made into the following form:

$$\mathbf{t}(\alpha, \beta) = t^{\frac{\alpha + \beta - [\alpha + \beta]_4}{4}}, \tag{87}$$

where $[\alpha]_4$ denotes $\alpha$ modulo 4. Here $t \in \mathcal{A}$ is invariant under $\mathbb{Z}_4^{\mathbf{T}}$. Once this form of the cocycle is fixed, there is still a remaining coboundary in the following form:

$$\varepsilon(\alpha) = \prod_{j=0}^{\alpha-1} {}^{\mathbf{T}^j}\varepsilon, \quad \alpha \in \{1, 2, 3\}, \tag{88}$$

where $\epsilon \in \mathcal{A}$. Under this coboundary, $t$ becomes $t \cdot T(\epsilon)$ where $T(\epsilon) = \prod_{j=0}^{3} {}^{\mathbf{T}^j}\varepsilon$.

If the fractionalization class is modified by such a torsor 2-cocycle, the fractional quantum number for a $\mathbf{T}$-invariant anyon $a$ becomes

$$(\eta_a^{\mathbf{T}}) = \eta_a^{\mathbf{T}} M_{a, \mathbf{t}(\mathbf{T}^2, \mathbf{T}^2)} = \eta_a^{\mathbf{T}} M_{at}. \tag{89}$$

Let us prove that $M_{a, T(\varepsilon)} = 1$, so $(\eta_a^{\mathbf{T}})$ is indeed an invariant. Note that

$$M_{a, T(\varepsilon)} = M_{t, \varepsilon} M_{t, {}^{\mathbf{T}}\varepsilon} M_{t, {}^{\mathbf{T}^2}\varepsilon} M_{t, {}^{\mathbf{T}^3}\varepsilon}. \tag{90}$$

Because $a$ is invariant under $\mathbf{T}$, we have $M_{a, {}^{\mathbf{T}}\varepsilon} = M_{a, \varepsilon}^*, M_{a, {}^{\mathbf{T}^3}\varepsilon} = M_{a, {}^{\mathbf{T}^2}\varepsilon}^*$, therefore $M_{a, T(\varepsilon)}$ is positive. Because $a$ and $\varepsilon$ are Abelian, $M_{a, T(\varepsilon)}$ must be a phase factor and thus $M_{a, T(\varepsilon)} = 1$. Clearly, the same is true for $\lambda_a^{\mathbf{T}^2}$.

## 8.3 Anomaly classification: $\mathbb{Z}_4^{\mathbf{T}}$ SPTs in (3+1)D and $\mathcal{H}^4(\mathbb{Z}_4^{\mathbf{T}}, \mathrm{U}(1))$

In (3+1)D, SPTs with $\mathbb{Z}_4^{\mathbf{T}}$ symmetry have a $\mathbb{Z}_2 \times \mathbb{Z}_2$ classification. One of the non-trivial classes comes from the group cohomology classification and is associated with

$$\mathcal{H}^4[\mathbb{Z}_4^{\mathbf{T}}, \mathrm{U}(1)] = \mathbb{Z}_2. \tag{91}$$

The other non-trivial SPT that is outside of the group cohomology classification is similar to the beyond group cohomology SPT state for $\mathbb{Z}_2^{\mathbf{T}}$.

We now define an invariant for the cohomology classes. It turns out that one can basically use the same formula for the $\mathcal{I}_3$ invariant of the $\mathbb{Z}_2 \times \mathbb{Z}_2^{\mathbf{T}}$ symmetry group discussed in Sec. 7, replacing $U_\pi$ with $\mathbf{T}^2$:

$$\mathcal{I} = \frac{\mathfrak{O}_r(\mathbf{T}, \mathbf{T}^2, \mathbf{T}^2, \mathbf{T}^2) \mathfrak{O}_r(\mathbf{T}^2, \mathbf{T}^2, \mathbf{T}, \mathbf{T}^2)}{\mathfrak{O}_r(\mathbf{T}^2, \mathbf{T}, \mathbf{T}^2, \mathbf{T}^2) \mathfrak{O}_r(\mathbf{T}^2, \mathbf{T}^2, \mathbf{T}^2, \mathbf{T})} \mathfrak{O}_r(\mathbf{T}^2, \mathbf{T}^2, \mathbf{T}^2, \mathbf{T}^2). \tag{92}$$

It is straightforward to check that this is invariant under any residual gauge transformations once we fix the canonical gauge where $\mathfrak{O}_r(\mathbf{g}_1, \mathbf{g}_2, \mathbf{g}_3, \mathbf{g}_4) = 1$ if any $\mathbf{g}_i = 1$. Recall that in this canonical gauge, residual gauge transformations correspond to shifting $\mathfrak{O}_r$ by a coboundary $d\epsilon$, where $\epsilon(\mathbf{g}_1, \mathbf{g}_2, \mathbf{g}_3) = 1$ if any $\mathbf{g}_i = 1$.

To understand the physics, let us for a moment enlarge the symmetry group to $[\mathrm{U}(1) \rtimes \mathbb{Z}_4^{\mathbf{T}}]/\mathbb{Z}_2$. The quotient means that the unitary $\mathbb{Z}_2$ element of $\mathbb{Z}_4^{\mathbf{T}}$ is identified with $U_\pi$. This is the symmetry group of charge-conserving "spin-1/2" bosons, i.e. a charge-1 boson carries $\mathbf{T}^2 = -1$. If the $\mathrm{U}(1)$ is broken down to the $\mathbb{Z}_2$ subgroup the group becomes just $\mathbb{Z}_4^{\mathbf{T}}$. The classification of such SPT phases can be understood through the property of (background) $\mathrm{U}(1)$ magnetic monopoles. Notice that the time-reversal transformation reverses magnetic charge. The nontrivial SPT phase is characterized by a topological theta term with $\Theta = 2\pi$, similar to the mixed anomaly of $\mathrm{U}(1) \rtimes \mathbb{Z}_2^{\mathbf{T}}$ symmetry [17]. Therefore the bulk also can be viewed as proliferating time-reversal domain walls decorated by (2+1)D BIQH with $\sigma_{xy} = 2$, and can be detected by the invariant Eq. (92) similar to $\mathcal{I}_3$.

## 8.4 General $\mathbb{Z}_4^{\mathbf{T}}$ anomaly formula

Using Eq. (87) and (92), a direct computation of the invariant gives

$$\mathcal{I} = \theta_t \eta_t^{\mathbf{T}}. \tag{93}$$

Note that $t = \mathbf{t}(\mathbf{T}^2, \mathbf{T}^2)$ can be interpreted as a "relative vison," in analogy to the cases with $\mathrm{U}(1)$ symmetry studied in Sec. 7. Thus the relative anomaly for $\mathbb{Z}_4^{\mathbf{T}}$ is non-trivial if either the relative vison is a fermion or if it carries fractional $\mathbb{Z}_2$ charge under $\mathbf{T}^2$ (in the reference SET), but not both.

## 8.5 Examples

### 8.5.1 $\mathcal{A} = \mathbb{Z}_2$ with $\mathbb{Z}_4^{\mathbf{T}}$ symmetry

Here the Abelian anyon sector associated with $\mathcal{A}$ consists of just two particles $\{1, x\}$. There are three possible fusion/braiding structures for such a braided fusion category:

1. $x$ is a boson. In this case, fusion and braiding are completely trivial.

2. $x$ is a fermion. We have $R^{x,x} = \theta_x = -1$.

3. $x$ is a semion/anti-semion, i.e. $\theta_x = \pm i$. Such a theory necessarily breaks time-reversal symmetry, so we do not consider this possibility.

There are two symmetry fractionalization classes. One can be related to another by $t = x$. Using Eq. (93), we find that the invariant for the relative anomaly is

$$\mathcal{I} = \theta_x \eta_x^{\mathbf{T}}. \tag{94}$$

Many theories fall into this class, including $D(S_3)$ (the quantum double of $S_3$, the permutation group on three elements) and $USp(4)_2$. See Refs. [21, 32] for a discussion of $\mathbb{Z}_2^{\mathbf{T}}$ time-reversal symmetry for these theories.

### 8.5.2 $\mathbb{Z}_N$ toric code with $\mathbb{Z}_4^{\mathbf{T}}$ symmetry

Let us consider the $\mathbb{Z}_N$ toric code with $\mathbb{Z}_4^{\mathbf{T}}$ symmetry. The anyons are labeled by $a = (a_1, a_2)$, for $a_i = 0, \cdots, N-1$, with fusion rules $(a_1, a_2) \times (b_1, b_2) = (a_1 + b_1, a_2 + b_2)$, modulo $N$. The $F$ symbols can all be chosen to be 1, unless they are not allowed by the fusion rules. The anyons have topological twist $\theta_{(a_1, a_2)} = e^{\frac{2\pi i}{N} a_1 a_2}$. We will take $R^{ab} = e^{\frac{2\pi i}{N} a_2 b_1}$. Therefore, under time-reversal $\mathbf{T}$, we must have that either $(a_1, a_2) \to (a_1, -a_2)$, or $(a_1, a_2) \to (a_2, -a_1)$. Note that the latter one squares to $(a_1, a_2) \to (-a_1, -a_2)$, which is a nontrivial operation for any $N > 2$. We consider the two cases separately.

*Case 1:* $\rho_{\mathbf{T}}(a_1, a_2) = (a_1, -a_2)$.

First we need to know $U_{\mathbf{g}}$ and $\eta$ in this case. Since $F$ symbols are all 1, and the $R$ symbols satisfy $R^{\mathbf{T}a\mathbf{T}b} = (R^{ab})^*$, we can pick all $U = 1$. Moreover, this means that there is a fractionalization class where we can set all $\eta = 1$ as well.

The symmetry fractionalization classification in this case is

$$\mathcal{H}_\rho^2[\mathbb{Z}_4, \mathbb{Z}_N \times \mathbb{Z}_N] = \mathbb{Z}_{(N,4)} \times \mathbb{Z}_{(N,2)}, \tag{95}$$

where in the above equation $(N, 4)$ means the greatest common divisor of $N$ and 4, and similarly for $(N, 2)$. To see this, first consider $N$ even. Since $t$ should be invariant under $\mathbb{Z}_4^{\mathbf{T}}$, we can write $t = (p, 0)$ or $t = (p, \frac{N}{2})$. The remaining coboundary takes the form of $(4k, 0)$. In other words, $p$ and $p + 4$ represent the same cohomology class. More generally, $p$ and $p + \gcd(N, 4)$ are the same. So the symmetry fractionalization classification is $\mathbb{Z}_{(N,4)} \times \mathbb{Z}_2$. For $N$ odd, $t$ must take the form $(p, 0)$ and because $(N, 4) = 1$, all of them are trivial.

Picking the reference state where all $\eta = 1$, the anomaly relative to this reference state becomes:

$$\mathcal{I} = \theta_t = \begin{cases} 1 & t = (p, 0) \\ (-1)^p & t = (p, N/2). \end{cases} \tag{96}$$

Note that the way to think about the anomalous case is that the $(1, 0)$ and $(0, N/2)$ particles carry half charge under $\mathbf{T}^2$. That is, $\eta_{(1,0)}^{\mathbf{T}} = \eta_{(0,N/2)}^{\mathbf{T}} = -1$, which makes it anomalous. To see this, we compute the fractional $\mathbf{T}^2$ charge for $t = (p, N/2)$ in the new theory: $\eta_{(1,0)}^{\mathbf{T}} = -1, \eta_{(0,N/2)}^{\mathbf{T}} = (-1)^p$, where recall that the shift in $\eta_a(\mathbf{g}, \mathbf{h})$ between the old and new theory is given by the mutual braiding phase $M_{a,t(\mathbf{g},\mathbf{h})}$.

This has an analog for $\mathbb{Z}_2^{\mathbf{T}}$ symmetry. Consider $\mathbb{Z}_N$ toric code with $\mathbb{Z}_2^{\mathbf{T}}$ symmetry, where $N$ is even. If we set $\eta_{(1,0)}^{\mathbf{T}} = -1$ and $\eta_{(0,N/2)}^{\mathbf{T}} = -1$, then $\mathcal{Z}(\mathbb{RP}^4) = -1$ [21], which is a generalization of the eTmT state [17] to the $\mathbb{Z}_N$ toric code. We can think of the generalization to $\mathbb{Z}_4^{\mathbf{T}}$ symmetry as the anomalous $eT^2mT^2$ state.

$$\text{\emph{Case 2:} } \rho_{\mathbf{T}}(a_1, a_2) = (a_2, -a_1).$$

Here we consider the case where $\rho$ is such that under time-reversal, $(a_1, a_2) \to (a_2, -a_1)$. We find the following expressions for $U$:

$$U_{\mathbf{T}}(a, b) = e^{\frac{2\pi i}{N} a_2 b_1}, \ U_{\mathbf{T}^2}(a, b) = e^{\frac{2\pi i}{N}(a_1 b_2 + a_2 b_1)}, \ U_{\mathbf{T}^3}(a, b) = e^{\frac{2\pi i}{N}(a_1 b_2 + 2a_2 b_1)}, \tag{97}$$

such that $\kappa_{\mathbf{g},\mathbf{h}}(a, b) = 1$. Therefore, one can set $\eta = 1$ as the reference class.

This is a special case of the example in Sec. 8.5.5, so we will skip it here.

### 8.5.3 Doubled semion with $\mathbb{Z}_4^{\mathbf{T}}$ symmetry

The doubled semion topological order has four anyon types: $\{1, s, s', ss'\}$, with $s^2 = s'^2 = 1$. The non-trivial $F$-symbols are $F_s^{sss} = -1$ and $F_{s'}^{s's's'} = -1$, while $R_1^{ss} = i$ and $R_1^{s's'} = -i$. Note that here there is no freedom in the action $\rho$ because $\mathbf{T}$ must necessarily interchange the two semions $s$ and $s'$. It is clear then that we can set all $U = 1$. Therefore, in the trivial fractionalization class, we can set all $\eta = 1$.

This is an interesting example, because with $\mathbb{Z}_2^{\mathbf{T}}$, we only have one possible symmetry fractionalization class: $\mathcal{H}_\rho^2[\mathbb{Z}_2, \mathbb{Z}_2 \times \mathbb{Z}_2] = \mathbb{Z}_1$. On the other hand,

$$\mathcal{H}_\rho^2[\mathbb{Z}_4, \mathbb{Z}_2 \times \mathbb{Z}_2] = \mathbb{Z}_2. \tag{98}$$

These two symmetry fractionalization classes are distinguished by whether the semions carry fractional or integer charge under $\mathbf{T}^2$. To see this, note that the non-trivial cocycle is given by $t = ss'$, which is the only $\mathbf{T}$-invariant anyon. The torsor 2-cocycle $t = ss'$ does not change $\eta_{ss'}^{\mathbf{T}}$, however it does change $\lambda_s^{\mathbf{T}^2}$ and $\lambda_s^{\mathbf{T}^2}$ by a sign.

We thus find that the anomaly vanishes:

$$\mathcal{I} = \theta_t = 1. \tag{99}$$

So both fractionalization classes can be realized in (2+1)D.

### 8.5.4 $\mathbb{Z}_N^{(p)}$ anyons

Let us consider $\mathbb{Z}_N^{(p)}$ anyons, where $N$ is an odd integer. The quasiparticles can be labeled $[a]$ for $a = 0, \cdots, N-1$. The $F$ and $R$ symbols are given by

$$F^{[a][b][c]} = 1, \ R_{[a+b]}^{[a][b]} = e^{i\frac{2\pi}{N} pab}. \tag{100}$$

Let us consider a general automorphism of the $\mathbb{Z}_N$ fusion group, $\mathbf{T} : [a] \to [ka]$, which requires $(k, N) = 1$. For the automorphism to correspond to an anti-unitary symmetry, we need $k^2 \equiv -1 \pmod{N}$. Then $\mathbf{T}^2 : [a] \to [-a]$, so $\mathbf{T}^4 = \mathbf{1}$, so that $\mathbf{T}$ generates a $\mathbb{Z}_4$ group. The condition $\theta_{[ka]} = \theta_{[a]}^*$, reduces to $p(k^2 + 1) \equiv 0 \pmod{N}$, which is obviously satisfied. Since there are no invariant anyons except $[0]$, the fractionalization classification is trivial, and therefore there is no relative anomaly to speak of.

### 8.5.5 U(1)×U(1) Chern-Simons theory

In this example we consider a U(1) × U(1) Chern-Simons theory:

$$\mathcal{L} = \frac{1}{4\pi} \varepsilon^{\mu\nu\lambda} a_\mu^I K_{IJ} \partial_\nu a_\lambda^J, \tag{101}$$

where $a^I, I = 1, 2$ are U(1) gauge fields. Here the integer K matrix is given by

$$K = \begin{pmatrix} m & n \\ n & -m \end{pmatrix}. \tag{102}$$

We will assume that $m$ is even so the theory is bosonic. The theory has a $\mathbb{Z}_4^{\mathbf{T}}$ symmetry generated by the following transformation):

$$\begin{pmatrix} a^1 \\ a^2 \end{pmatrix} \to \begin{pmatrix} 0 & 1 \\ -1 & 0 \end{pmatrix} \begin{pmatrix} a^1 \\ a^2 \end{pmatrix}. \tag{103}$$

Quasiparticles in the theory are labeled by their gauge charges, in this case a two-dimensional integer column vector $\mathbf{l} = (l_1, l_2)^{\mathsf{T}}$. Local excitations all take the form $K\mathbf{l}'$ for some integer vector $\mathbf{l}'$, so anyon types are defined by equivalence classes $\mathbf{l} \sim \mathbf{l} + K\mathbf{l}'$. Notice that $\mathbf{T}^2$ sends $\mathbf{l}$ to $-\mathbf{l}$, so in general $\mathbf{T}$ is of order 4, unless all anyons are self dual. This includes the case 2 of the $\mathbb{Z}_N$ toric code example mentioned above as a special case with $m = 0$. Another interesting case is that when $m = n$, the $K$ matrix is SL$(2, \mathbb{Z})$ equivalent to $\begin{pmatrix} 2n & \\ & -n \end{pmatrix}$. One can show that the minimal time-reversal symmetry in this case is of order 4.

Let specialize to the case where $n$ is even. We show in Appendix A that for the $\mathbb{Z}_4^{\mathbf{T}}$ symmetry, while the $U$ symbols are nontrivial the $\kappa_{\mathbf{g,h}}$ symbols can all be set to 1, for the case where $m$ and $n$ are both even. Thus in this case there is a reference state with $\eta = 1$. To determine fractionalization classes, let us find all $\mathbf{T}$-invariant anyons. With a little algebra, we find that for even $n$ there is a unique nontrivial $\mathbf{T}$-invariant anyon, given by the following charge vector:

$$\begin{pmatrix} \frac{m+n}{2} \\ \frac{n-m}{2} \end{pmatrix}. \tag{104}$$

To see what distinguishes the two classes related by $t = (\frac{m+n}{2}, \frac{n-m}{2})^{\mathsf{T}}$, we need to compute fractional quantum numbers. Since $\mathbf{T}^2$ is just charge conjugation, according to Eq. (107), we may associate an invariant $\tilde{\lambda}_a^{\mathbf{T}^2} = \pm 1$ that changes sign when the symmetry fractionalization class is changed from $a$ carrying fractional $\mathbf{T}^2$ charge to integer $\mathbf{T}^2$ charge. The fractionalization class changes $\tilde{\lambda}_{(x,y)}^{\mathbf{T}^2} \to \tilde{\lambda}_{(x,y)}^{\mathbf{T}^2}(-1)^{x+y}$ for a quasiparticle $(x, y)^{\mathsf{T}}$.

Note however that $\eta_t^{\mathbf{T}} = (-1)^n = 1$. Then relative to the trivial reference SET phase we find the cohomology invariant for the anomaly is

$$\mathcal{I} = \theta_t = (-1)^{n/2}. \tag{105}$$

# 9 $\mathbb{Z}_2 \times \mathbb{Z}_2$ symmetry

In this section we consider a unitary symmetry $G = \mathbb{Z}_2 \times \mathbb{Z}_2 = \{\mathbf{1}, \mathbf{Z}, \mathbf{X}, \mathbf{Y} = \mathbf{ZX}\}$.

## 9.1 Symmetry fractionalization for $\mathbb{Z}_2 \times \mathbb{Z}_2$

There are three $\mathbb{Z}_2$ subgroups of $G$, generated by $\mathbf{X}, \mathbf{Y}$ and $\mathbf{Z}$ respectively. $\mathbb{Z}_2 \times \mathbb{Z}_2$ symmetry fractionalization in a topological phase can be characterized by fractional charges under these $\mathbb{Z}_2$ subgroups.

### 9.1.1 Fractional $\mathbb{Z}_2$ charge

Let us first focus on a single $\mathbb{Z}_2$ subgroup, and denote the generator by $\mathbf{g}$. We now define two types of invariants to characterize $\mathbb{Z}_2$ fractionalization of an anyon $a$.

**Type-I** Consider the case where $a = {}^\mathbf{g}a$, and the order of $a$ is even (here order means the minimal integer $n$ such that $a^n \to 1$). We further assume that at least one of the fusion tree basis states for $a \times a \times \cdots a \to 1$ is $\mathbb{Z}_2$ invariant. Namely, we can find a series of $\mathbf{g}$-invariant anyons $a_0 = a, a_1, a_2, \cdots, a_{n-2}, a_{n-1} = 1$, such that $N_{a,a_j}^{a_{j+1}} > 0$ for $j = 0, 1, \ldots, n-2$. Let us define

$$\lambda_a^\mathbf{g} = \eta_a^{n/2}(\mathbf{g}, \mathbf{g}) \prod_{j=0}^{n-2} U_\mathbf{g}(a, a_j; a_{j+1}). \tag{106}$$

The product of $U$'s is simply the action of $\rho_\mathbf{g}$ on a state corresponding to the fusion tree $a \times a \times \cdots a \to 1$. Intuitively, $\lambda_a^\mathbf{g} = -1$ means that fusing $n$ identical copies of $a$ yields a $\mathbb{Z}_2$ charge. One can show that $(\lambda_a^\mathbf{g})^2 = 1$ and the invariant $\lambda_a^\mathbf{g} = \pm 1$ determines whether $a$ carries $\mathbb{Z}_2$ fractional charge.

**Type-II** When $\bar{a} = {}^\mathbf{g}a$, we define

$$\tilde{\lambda}_a^\mathbf{g} = \eta_a(\mathbf{g}, \mathbf{g}) U_\mathbf{g}(a, \bar{a}; 1) R_1^{\bar{a}, a} \theta_a. \tag{107}$$

The $R$ symbol is introduced so that $\tilde{\lambda}_a^\mathbf{g}$ is invariant under (vertex basis) gauge transformations. $\theta_a$ is introduced so that $(\tilde{\lambda}_a^\mathbf{g})^2 = 1$. This invariant is most easily understood when $\mathbf{g}$ corresponds to a spatial rotation by $\pi$, i.e. inversion. In this case, one may create a $\mathbf{g}$-invariant physical state by placing $a$ and $\bar{a}$ in inversion-symmetric positions, and $\tilde{\lambda}_a^\mathbf{g}$ is the eigenvalue of inversion acting on this state. We caution that $\tilde{\lambda}_a^\mathbf{g}$ is best thought of as a relative invariant; there are examples where $\tilde{\lambda}_a^\mathbf{g} = -1$ but yet $\mathbb{Z}_2$ symmetry is not fractionalized (namely the case of $U(1)_4$ with charge conjugation symmetry), and there are examples where $\tilde{\lambda}_a^\mathbf{g} = 1$ and $\mathbb{Z}_2$ symmetry is fractionalized. Nevertheless, $\tilde{\lambda}_a^\mathbf{g} = \pm 1$ is a gauge invariant quantity and changes value when the $\mathbb{Z}_2$ symmetry fractionalization class is changed.

Let us now make a connection with the symmetry fractionalization classification. Choose a cohomology class $[\mathbf{t}] \in \mathcal{H}_\rho^2[G, \mathcal{A}]$. We use the canonical gauge $\mathbf{t}(1, \mathbf{g}) = \mathbf{t}(\mathbf{g}, 1) = 1$. The 2-cocycle condition for $\mathbf{t}$ reads

$$^\mathbf{g}\mathbf{t}(\mathbf{g}, \mathbf{g}) = \mathbf{t}(\mathbf{g}, \mathbf{g}). \tag{108}$$

In the following we will simply write $\mathbf{t}$ for $\mathbf{t}(\mathbf{g}, \mathbf{g})$. There is a coboundary $\mathbf{t}(\mathbf{g}, \mathbf{g}) \to \mathbf{t}(\mathbf{g}, \mathbf{g}) \times \varepsilon \times {}^\mathbf{g}\varepsilon$.

Upon changing the symmetry fractionalization class by $[\mathbf{t}]$, the $\eta$ symbols change to

$$\eta_a'(\mathbf{g}, \mathbf{g}) = \eta_a(\mathbf{g}, \mathbf{g}) M_{a, \mathbf{t}}. \tag{109}$$

One can also verify that, as expected, changing the symmetry fractionalization class by $[\mathbf{t}]$ can only change the $\lambda^\mathbf{g}$ and $\tilde{\lambda}^\mathbf{g}$ invariants by $\pm 1$. For type-I, $\lambda_a^{\mathbf{g}\prime} = \lambda_a^\mathbf{g} M_{a, \mathbf{t}}^{n/2}$. It is obvious that $M_{a, \mathbf{t}}^{n/2} = \pm 1$. For type-II, $\tilde{\lambda}_a^{\mathbf{g}\prime} = \tilde{\lambda}_a^\mathbf{g} M_{a, \mathbf{t}}$. One can see that $M_{a, \mathbf{t}} = \pm 1$ as follows:

$$M_{a, \mathbf{t}} = M_{\bar{a}, \mathbf{t}}^* = M_{\mathbf{g}a, \mathbf{t}}^* = M_{a, {}^\mathbf{g}\mathbf{t}}^* = M_{a, \mathbf{t}}^*. \tag{110}$$

Thus $M_{a, \mathbf{t}}$ is real. Since $\mathbf{t}$ is Abelian, it follows that $M_{a, \mathbf{t}} = \pm 1$.

### 9.1.2 $\mathbb{Z}_2 \times \mathbb{Z}_2$ fractionalization classes

We now give an explicit description of 2-cocycles in $\mathcal{H}_{[\rho]}^2[\mathbb{Z}_2 \times \mathbb{Z}_2, \mathcal{A}]$. We fix a gauge such that $\mathbf{t}(\mathbf{X}, \mathbf{Z}) = 1$. Then by systematically solving the 2-cocycle conditions, one can show that all 2-cocycles can be expressed in terms of $\mathbf{t}(\mathbf{X}, \mathbf{X}), \mathbf{t}(\mathbf{Z}, \mathbf{Z})$ and $\mathbf{t}(\mathbf{Y}, \mathbf{Y})$. They have to satisfy $^\mathbf{g}\mathbf{t}(\mathbf{g}, \mathbf{g}) = \mathbf{t}(\mathbf{g}, \mathbf{g})$ for $\mathbf{g} = \mathbf{X}, \mathbf{Y}, \mathbf{Z}$, and

$$^\mathbf{Z}\mathbf{t}(\mathbf{Y}, \mathbf{Y}) \times \mathbf{t}(\mathbf{Y}, \mathbf{Y}) = {}^\mathbf{X}\mathbf{t}(\mathbf{Z}, \mathbf{Z}) \times \mathbf{t}(\mathbf{Z}, \mathbf{Z}) \times {}^\mathbf{Y}\mathbf{t}(\mathbf{X}, \mathbf{X}) \times \mathbf{t}(\mathbf{X}, \mathbf{X}). \tag{111}$$

Finally, they are subject to coboundaries $\mathbf{t}(\mathbf{g},\mathbf{g}) \to \mathbf{t}(\mathbf{g},\mathbf{g}) \times \varepsilon(\mathbf{g}) \times^{\mathbf{g}} \varepsilon(\mathbf{g})$, with $\varepsilon(\mathbf{Y}) = \varepsilon(\mathbf{X}) \times^{\mathbf{X}} \varepsilon(\mathbf{Z})$.

Another result that is useful for the examples that we study is:

$$t(\mathbf{Z},\mathbf{X}) = \frac{^{\mathbf{X}}\mathbf{t}(\mathbf{Y},\mathbf{Y})}{\mathbf{t}(\mathbf{X},\mathbf{X})^{\mathbf{X}}\mathbf{t}(\mathbf{Z},\mathbf{Z})}. \tag{112}$$

## 9.2 Anomaly classification: $\mathbb{Z}_2 \times \mathbb{Z}_2$ SPTs in (3+1)D

(3+1)D SPTs with $\mathbb{Z}_2 \times \mathbb{Z}_2$ symmetry are classified by

$$\mathcal{H}^4[\mathbb{Z}_2 \times \mathbb{Z}_2, \mathrm{U}(1)] = \mathbb{Z}_2^2. \tag{113}$$

We can define two $\mathbb{Z}_2$ invariants [37]:

$$\begin{aligned} \mathcal{I}_{X,Z} &= \chi_{\mathbf{Z}}(\mathbf{X},\mathbf{X},\mathbf{X}), \\ \mathcal{I}_{Z,X} &= \chi_{\mathbf{X}}(\mathbf{Z},\mathbf{Z},\mathbf{Z}), \end{aligned} \tag{114}$$

where $\chi$ is the slant product:

$$\chi_{\mathbf{h}}(\mathbf{g},\mathbf{g},\mathbf{g}) = \frac{\mathcal{O}_r(\mathbf{g},\mathbf{h},\mathbf{g},\mathbf{g})\mathcal{O}_r(\mathbf{g},\mathbf{g},\mathbf{g},\mathbf{h})}{\mathcal{O}_r(\mathbf{h},\mathbf{g},\mathbf{g},\mathbf{g})\mathcal{O}_r(\mathbf{g},\mathbf{g},\mathbf{h},\mathbf{g})}. \tag{115}$$

## 9.3 Examples with permutations

Anomalies for non-permuting Abelian unitary symmetries and Abelian topological orders were thoroughly studied in Ref. [38]. Instead here we study two examples with anyon-permuting $\mathbb{Z}_2 \times \mathbb{Z}_2$ symmetry.

### 9.3.1 $\mathbb{Z}_N$ toric code

We consider $\mathbb{Z}_N$ toric code with even $N$, whose topological symmetry group always contains a $\mathbb{Z}_2 \times \mathbb{Z}_2$ subgroup, generated by electromagnetic duality $(a_1, a_2) \to (a_2, a_1)$ and charge conjugation $C : (a_1, a_2) \to (N-a_1, N-a_2)$. Denote by $\mathcal{A}_C = \{(0,0), (N/2,0), (0,N/2), (N/2,N/2)\}$ the set of self-dual anyons.

We consider the case where $\rho_{\mathbf{X}} = C, \rho_{\mathbf{Z}} = \mathbb{1}$. We then have $\mathbf{t}(\mathbf{X},\mathbf{X}), \mathbf{t}(\mathbf{Y},\mathbf{Y}) \in \mathcal{A}_C$, both of which are gauge-invariant. Accounting for the gauge freedom for $\mathbf{t}(\mathbf{Z},\mathbf{Z})$, we have $\mathbf{t}(\mathbf{Z},\mathbf{Z}) \in \{(0,0),(1,0),(0,1),(1,1)\}$. We will denote $t_{\mathbf{g}} \equiv \mathbf{t}(\mathbf{g},\mathbf{g})$.

For the case $\rho_{\mathbf{X}} = C$, $\rho_{\mathbf{Z}} = \mathbb{1}$, one can show that all U symbols can be set to 1. There is therefore a reference state where all $\eta$ can be set to 1. The $\mathbb{Z}_2$ fractional charges can be found to be

$$\tilde{\lambda}_a^{\mathbf{g}} = M_{a,t_{\mathbf{g}}}, \quad \mathbf{g} = \mathbf{X}, \mathbf{Y}, \tag{116}$$

$$\lambda_a^{\mathbf{Z}} = M_{a,t_{\mathbf{Z}}}^{N/2}. \tag{117}$$

The obstruction formula relative to this reference state then just depends on the $R$ symbol. The invariants are found to be

$$\begin{aligned} \mathcal{I}_{\mathbf{Z},\mathbf{X}} &= M_{t_Z,t_X} M_{t_Z,t_Y}, \\ \mathcal{I}_{\mathbf{X},\mathbf{Z}} &= M_{t_X,t_Z} M_{t_X,t_Y}. \end{aligned} \tag{118}$$

Table 1: Obstruction classes for $\mathbb{Z}_2 \times \mathbb{Z}_2$ symmetry in $U(1)_{2N}$ with even $N$.

| $\mathbf{t(X,X)}$ | $\mathbf{t^N(Z,Z)}$ | $\mathbf{t(Y,Y)}$ | $\tilde{\lambda}^{\mathbf{X}}_{[1]}, \tilde{\lambda}^{\mathbf{Z}}_{[1]}, \tilde{\lambda}^{\mathbf{Y}}_{[1]}$ | $\mathcal{I}_{\mathbf{X},Z}$ | $\mathcal{I}_{\mathbf{Z},\mathbf{X}}$ |
|:---:|:---:|:---:|:---:|:---:|:---:|
| 0 | 0 | 0 | $1, 1, 1$ | 1 | 1 |
| 0 | 0 | $[N]$ | $1, 1, -1$ | 1 | 1 |
| 0 | $[N]$ | 0 | $1, -1, 1$ | 1 | 1 |
| 0 | $[N]$ | $[N]$ | $1, -1, -1$ | 1 | $-1$ |
| $[N]$ | 0 | 0 | $-1, 1, 1$ | 1 | 1 |
| $[N]$ | 0 | $[N]$ | $-1, 1, -1$ | 1 | 1 |
| $[N]$ | $[N]$ | 0 | $-1, -1, 1$ | $-1$ | $-1$ |
| $[N]$ | $[N]$ | $[N]$ | $-1, -1, -1$ | $-1$ | 1 |

### 9.3.2   $U(1)_{2N}$ **Chern-Simons theory**

We consider $U(1)_{2N}$ Chern-Simons theory, whose topological symmetry group always contains a $\mathbb{Z}_2$ charge conjugation symmetry.

Anyons in this case are labeled by $a = 0, 1, \ldots, 2N-1$ defined mod $2N$. The $F$ and $R$ symbols read:

$$F^{abc} = e^{\frac{i\pi}{2N}a(b+c-[b+c])},$$
$$R^{ab} = e^{\frac{i\pi}{2N}ab}, \tag{119}$$

where $[a] = a \bmod 2N$. The $\mathbb{Z}_2$ charge conjugation symmetry has the action $C : a \rightarrow 2N - a$. The corresponding $U$ symbols are found to be

$$U_C(a, b) = \begin{cases} (-1)^a & b > 0 \\ 1 & b = 0. \end{cases} \tag{120}$$

Again consider $G = \mathbb{Z}_2 \times \mathbb{Z}_2$, with $\rho_{\mathbf{X}} = C, \rho_{\mathbf{Z}} = \mathbb{1}$. It is straightforward to check that $\kappa_{\mathbf{g},\mathbf{h}}(a, b) = 1$, so there is a canonical reference state with $\eta = 1$. Adopting the results in Sec. 9.1.2, the 2-cocycles are labeled by $\mathbf{t(X,X)}, \mathbf{t(Y,Y)}$ and $\mathbf{t^N(Z,Z)}$ (we raise $\mathbf{t(Z,Z)}$ to the $N$-th power to eliminate remaining coboundary degrees of freedom). They are all valued in $\{[0], [N]\}$, and subject to no further constraints. We notice that when $N$ is odd, the MTC can be factorized into $\mathbb{Z}_2^{(\frac{N}{2})} \times \mathbb{Z}_N^{(1)}$, where $\mathbb{Z}_2^{(\frac{N}{2})} = \{[0], [N]\}$ is a semion/anti-semion theory and $\mathbb{Z}_N^{(1)} = \{[0], [2], [4], \ldots, [2N]\}$, and all symmetry fractionalizations can be accounted for entirely in the semion sector, which was treated extensively in Refs. [18, 20]. So we will only present the results for even $N$. The corresponding anomaly invariants are listed in Table 1.

# 10   $\mathbb{Z}_2 \times \mathbb{Z}_2^{\mathbf{T}}$ **symmetry**

The case of $\mathbb{Z}_2 \times \mathbb{Z}_2^{\mathbf{T}}$ symmetry is closely related to the case of $U(1) \times \mathbb{Z}_2^{\mathbf{T}}$ symmetry, studied in Sec. 9.1. However, as we see below, the possible symmetry fractionalization classes are richer, which leads to new possibilities.

We denote the group as $\{1, \mathbf{X}, \mathbf{T}, \mathbf{T}' = \mathbf{XT}\}$ where $\mathbf{X}$ generates the $\mathbb{Z}_2$ subgroup and $\mathbf{T}$ generates the $\mathbb{Z}_2^{\mathbf{T}}$ subgroup.

## 10.1 $\mathbb{Z}_2 \times \mathbb{Z}_2^{\mathbf{T}}$ SPTs in (1+1)D

In (1+1)D, $\mathbb{Z}_m \times \mathbb{Z}_2^{\mathbf{T}}$ SPTs have a classification given by

$$\mathcal{H}^2[\mathbb{Z}_m \times \mathbb{Z}_2^{\mathbf{T}}, \mathrm{U}(1)] = \mathbb{Z}_2 \times \mathbb{Z}_{(m,2)}, \tag{121}$$

where $(m, 2)$ means the greatest common divisor of $m$ and 2. Thus $\mathcal{H}^2[\mathbb{Z}_2 \times \mathbb{Z}_2^{\mathbf{T}}, \mathrm{U}(1)] = \mathbb{Z}_2 \times \mathbb{Z}_2$. The two $\mathbb{Z}_2$ classes correspond to whether the edge modes transform projectively as $\mathbf{T}^2 = \pm 1$ and $(\mathbf{T}')^2 = \pm 1$.

## 10.2 Symmetry fractionalization for $\mathbb{Z}_2 \times \mathbb{Z}_2^{\mathbf{T}}$

Symmetry fractionalization for $\mathbb{Z}_2 \times \mathbb{Z}_2^{\mathbf{T}}$ is classified by elements of $\mathcal{H}^2_{[\rho]}[\mathbb{Z}_2 \times \mathbb{Z}_2, \mathcal{A}]$. As discussed in Section 9.1, elements of this group are completely parametrized by $\mathfrak{t}(\mathbf{g}, \mathbf{g})$, for $\mathbf{g} = \mathbf{X}, \mathbf{T}, \mathbf{XT}$, with the condition that $^{\mathbf{g}}\mathfrak{t}(\mathbf{g}, \mathbf{g}) = \mathfrak{t}(\mathbf{g}, \mathbf{g})$.

We see that the symmetry fractionalization classes in this case cannot be completely characterized in terms of projective representations of $\mathbb{Z}_2 \times \mathbb{Z}_2^{\mathbf{T}}$ or, equivalently, by dimensional reduction to (1+1)D. The additional information is $\mathfrak{t}(\mathbf{X}, \mathbf{X})$, i.e. the fractional $\mathbb{Z}_2$ charge.

## 10.3 $\mathbb{Z}_2 \times \mathbb{Z}_2^{\mathbf{T}}$ SPTs in (3+1) D

The classification of $\mathbb{Z}_2 \times \mathbb{Z}_2^{\mathbf{T}}$ SPTs in (3+1)D and thus the anomaly classification for (2+1)D SETs is identical to the case of $\mathrm{U}(1) \times \mathbb{Z}_2^{\mathbf{T}}$ SPTs. Namely, within group cohomology, there is a $\mathbb{Z}_2^3$ classification:

$$\mathcal{H}^4[\mathbb{Z}_2 \times \mathbb{Z}_2^{\mathbf{T}}, \mathrm{U}(1)] = \mathbb{Z}_2^3. \tag{122}$$

There is an additional $\mathbb{Z}_2$ associated with the beyond group cohomology pure $\mathbb{Z}_2^{\mathbf{T}}$ SPT. One of the $\mathbb{Z}_2$ factors within group cohomology is also associated with a pure $\mathbb{Z}_2^{\mathbf{T}}$ SPT state. Thus we have a $\mathbb{Z}_2^2$ classification coming from pure $\mathbb{Z}_2^{\mathbf{T}}$ SPTs, and an additional $\mathbb{Z}_2^2$ factor arising due to a mixing between the $\mathbb{Z}_2$ and $\mathbb{Z}_2^{\mathbf{T}}$ symmetries.

In terms of 4-cocycles, the invariants that describe the $\mathbb{Z}_2^3$ classification are identical to those discussed in Sec. 7.2 for the $\mathrm{U}(1) \times \mathbb{Z}_2^{\mathbf{T}}$ case, in Eq. (71), (72). We take $\mathbf{X} = U_\pi$ to be the generator of the $\mathbb{Z}_2$, and $\mathbf{T}' = \mathbf{XT}$.

## 10.4 Example: $\mathbb{Z}_2$ toric code

### 10.4.1 With no permutations

Here we study a simple example, the $\mathbb{Z}_2$ toric code state where the symmetries do not permute the particle types. In this case,

$$\mathcal{H}^2[\mathbb{Z}_2 \times \mathbb{Z}_2, \mathbb{Z}_2] = \mathbb{Z}_2^3. \tag{123}$$

In contrast recall that for $\mathrm{U}(1) \times \mathbb{Z}_2^{\mathbf{T}}$ symmetry, we have $\mathcal{H}^2[\mathrm{U}(1) \times \mathbb{Z}_2, \mathbb{Z}_2] = \mathbb{Z}_2^2$. The $\mathrm{U}(1) \times \mathbb{Z}_2^{\mathbf{T}}$ classes correspond to the specific case where $\mathfrak{t}(\mathbf{T}', \mathbf{T}') = \mathfrak{t}(\mathbf{X}, \mathbf{X})\mathfrak{t}(\mathbf{T}, \mathbf{T})$.

For the $\mathbb{Z}_2$ toric code, we thus have a total of 64 possible choices, since $\mathfrak{t}(\mathbf{g}, \mathbf{g}) = 1, e, m, \psi$, for $\mathbf{g} = \mathbf{X}, \mathbf{T}, \mathbf{T}'$. These can be physically understood as whether the $e$ (or $m$) particle carries charge $1/2$, and whether it carries Kramers degeneracy under $\mathbf{T}$ or $\mathbf{T}'$.

Recall that for the toric code the $F$-symbols are all either one or zero depending on whether they are allowed by the fusion rules. Furthermore, since $\rho$ is trivial in this example, we can pick a reference state where all $\eta$ and $U$ are set equal to 1. Thus, the relative anomaly is simply

$$\mho_r(\mathbf{g}, \mathbf{h}, \mathbf{k}, \mathbf{l}) = R^{\mathfrak{t}(\mathbf{k}, \mathbf{l}), \mathfrak{t}(\mathbf{g}, \mathbf{h})}. \tag{124}$$

We note that the form of the relative anomaly in $\mathbb{Z}_N$ toric code holds for any symmetry group as long as no anyons are permuted. This result was derived previously by explicitly constructing generalized string-net models on the surface of (3+1)d SPT phase in Ref. [28].

Thus the invariants are:

$$\begin{aligned}
\mathcal{I}_1 &= \mathfrak{O}_r(\mathbf{T}, \mathbf{T}, \mathbf{T}, \mathbf{T}) = \theta_{\mathbf{t}(\mathbf{T}, \mathbf{T})}, \\
\mathcal{I}_2 &= \mathfrak{O}_r(\mathbf{T}', \mathbf{T}', \mathbf{T}', \mathbf{T}') = \theta_{\mathbf{t}(\mathbf{T}', \mathbf{T}')},
\end{aligned} \tag{125}$$

and

$$\mathcal{I}_3 = R^{\mathbf{t}(\mathbf{X},\mathbf{X}),\mathbf{t}(\mathbf{T},\mathbf{X})}R^{\mathbf{t}(\mathbf{T},\mathbf{X}),\mathbf{t}(\mathbf{X},\mathbf{X})}\theta_{\mathbf{t}(\mathbf{X},\mathbf{X})} = M_{\mathbf{t}(\mathbf{X},\mathbf{X}),\mathbf{t}(\mathbf{T},\mathbf{T})}M_{\mathbf{t}(\mathbf{X},\mathbf{X}),\mathbf{t}(\mathbf{T}',\mathbf{T}')}\theta_{\mathbf{t}(\mathbf{X},\mathbf{X})}, \tag{126}$$

where we have used $M_{ab} = R^{ab}R^{ba}$. We have also used the gauge $\mathbf{t}(\mathbf{X}, \mathbf{T}) = 1$, with the cocycle condition in this case giving $\mathbf{t}(\mathbf{T}, \mathbf{X}) = \mathbf{t}(\mathbf{T}', \mathbf{T}')\mathbf{t}(\mathbf{X}, \mathbf{X})\mathbf{t}(\mathbf{T}, \mathbf{T})$.

Table 2 summarizes the anomalies for all of the 36 inequivalent symmetry fractionalization classes. (Of the 64 possible classes associated with $\mathcal{H}^2(\mathbb{Z}_2 \times \mathbb{Z}_2, \mathbb{Z}_2 \times \mathbb{Z}_2) = \mathbb{Z}_2^6$, relabeling $e$ and $m$ gives 36 inequivalent classes). Note that we use the labeling convention of Ref. [39]: If the $e$ particle carries half-charge under the $\mathbb{Z}_2$, it is followed by a $C$ in the labeling. If $e$ carries a Kramers degeneracy under $\mathbf{T}$ or $\mathbf{T}'$, then it is followed by a $T$ or $T'$ in the labeling.

One can consider $\mathbf{t}(\mathbf{X}, \mathbf{X})$ to correspond to the vison. From Table 2, we see that in general $\mathcal{I}_3$ is no longer determined by whether the vison is a fermion, in contrast to the case with $U(1) \times \mathbb{Z}_2^{\mathbf{T}}$ symmetry.

### 10.4.2 With permutations

The topological symmetry group of the $\mathbb{Z}_2$ toric code is $\mathbb{Z}_2$, with the nontrivial element being the "electromagnetic duality" that swaps $e$ with $m$.

First we compute the corresponding $U$ symbols for this symmetry. It is easy to see that in a gauge where all $F$ and $R$ symbols are real, we do not need to distinguish unitary and anti-unitary symmetries at least for $U$. One solution is

$$U(a, b) = (-1)^{a_2 b_1}, \tag{127}$$

which leads to

$$\kappa_{\mathbf{X},\mathbf{X}}(a, b) = (-1)^{a_2 b_1 + a_1 b_2} = \frac{\theta_a \theta_b}{\theta_{a \times b}}. \tag{128}$$

Next we consider constraints on $\eta$ symbols for a $\mathbb{Z}_2$ symmetry $\mathbf{g}$ which maps to the duality symmetry. The only nontrivial cocycle is $\eta_a(\mathbf{g}, \mathbf{g}) \equiv \eta_a$. The fusion rule implies that $\eta_e^2 = \eta_m^2 = \eta_\psi^2 = 1$, and $\eta_\psi = -\eta_e \eta_m$. Thus we find $\eta_a = \pm 1$ for all $a$. The twisted 2-cocycle condition implies $\eta_e \eta_m = 1$, for both unitary and anti-unitary $\mathbf{g}$. So we have found that $\eta_\psi = -1$.

Now if $\mathbf{g}$ is unitary, the $\mathbb{Z}_2$ fractional charge

$$\lambda_\psi^{\mathbf{g}} = \eta_\psi(\mathbf{g}, \mathbf{g})U_{\mathbf{g}}(\psi, \psi; 1) = 1. \tag{129}$$

The other values $\eta_e$ and $\eta_m$ can be set to 1 by gauge transformations.

If $\mathbf{g}$ is anti-unitary, $\eta_\psi$ is the gauge-invariant $\mathbf{T}^2$ value, so the fermion $\psi$ has $\mathbf{T}^2 = -1$, which is a well-known result.

In both cases, there are no further choices for symmetry fractionalization. This is also consistent with $\mathcal{H}_\rho^2[\mathbb{Z}_2, \mathbb{Z}_2 \times \mathbb{Z}_2] = \mathbb{Z}_1$ for this choice of $\rho$.

Table 2: Anomalies for $\mathbb{Z}_2$ toric code with $\mathbb{Z}_2 \times \mathbb{Z}_2^{\mathbf{T}}$ symmetry, where symmetries do not permute any particle types. e0m0 refers to the trivial symmetry fractionalization class. If e or m does not appear in the label, then it has trivial symmetry fractionalization quantum numbers.

| Label | $(\mathbf{t}(\mathbf{X},\mathbf{X}), \mathbf{t}(\mathbf{T},\mathbf{T}), \mathbf{t}(\mathbf{T}',\mathbf{T}'))$ | $(\mathcal{I}_1, \mathcal{I}_2, \mathcal{I}_3)$ |
|---|---|---|
| e0m0 | $(1,1,1)$ | (1,1,1) |
| eT$'$ | $(1,1,m)$ | (1, 1, 1) |
| eT$'$mT$'$ | $(1,1,\psi)$ | (1, -1, 1) |
| eT | $(1,m,1)$ | (1, 1, 1) |
| eTmT$'$ | $(1,m,e)$ | (1, 1, 1) |
| eTT$'$ | $(1,m,m)$ | (1, 1, 1) |
| eTT$'$mT$'$ | $(1,m,\psi)$ | (1, -1, 1) |
| eTmT | $(1,\psi,1)$ | (-1, 1, 1) |
| eTT$'$mT | $(1,\psi,m)$ | (-1, 1, 1) |
| eTT$'$mTT$'$ | $(1,\psi,\psi)$ | (-1, -1, 1) |
| mC | $(e,1,1)$ | (1, 1, 1) |
| mCT$'$ | $(e,1,e)$ | (1, 1, 1) |
| eT$'$mC | $(e,1,m)$ | (1, 1, -1) |
| eT$'$mCT$'$ | $(e,1,\psi)$ | (1, -1, -1) |
| mCT | $(e,e,1)$ | (1, 1, 1) |
| mCTT$'$ | $(e,e,e)$ | (1, 1, 1) |
| eT$'$mCT | $(e,e,m)$ | (1, 1, -1) |
| eT$'$mCTT$'$ | $(e,e,\psi)$ | (1, -1, -1) |
| eTmC | $(e,m,1)$ | (1, 1, -1) |
| eTmCT$'$ | $(e,m,e)$ | (1, 1, -1) |
| eTT$'$mC | $(e,m,m)$ | (1, 1, 1) |
| eTT$'$mCT$'$ | $(e,m,\psi)$ | (1, -1, 1) |
| eTmCT | $(e,\psi,1)$ | (-1, 1, -1) |
| eTmCTT$'$ | $(e,\psi,e)$ | (-1, 1, -1) |
| eTT$'$mCT | $(e,\psi,m)$ | (-1, 1, 1) |
| eTT$'$mCTT$'$ | $(e,\psi,\psi)$ | (-1, -1, 1) |
| eCmC | $(\psi,1,1)$ | (1, 1, -1) |
| eCT$'$mC | $(\psi,1,m)$ | (1, 1, 1) |
| eCT$'$mCT$'$ | $(\psi,1,\psi)$ | (1, -1, -1) |
| eCmCT | $(\psi,e,1)$ | (1, 1, 1) |
| eCmCTT$'$ | $(\psi,e,e)$ | (1, 1, -1) |
| eCT$'$mCT | $(\psi,e,m)$ | (1, 1, -1) |
| eCT$'$mCTT$'$ | $(\psi,e,\psi)$ | (1, -1, 1) |
| eCTmCT | $(\psi,\psi,1)$ | (-1, 1, -1) |
| eCTT$'$mCT | $(\psi,\psi,m)$ | (-1, 1, 1) |
| eCTT$'$mCTT$'$ | $(\psi,\psi,\psi)$ | (-1, -1, -1) |

- Consider the case where $\mathbf{X}$ permutes $e$ and $m$, and $\mathbf{T}$ does not. We can gauge fix $\mathbf{t}(\mathbf{X},\mathbf{X}) = \mathbf{t}(\mathbf{T}',\mathbf{T}') = 1$. In this case we find $\mathbf{t}(\mathbf{T},\mathbf{T}) = 1$ or $\psi$. We find that $\mathcal{I}_1 = \theta_{\mathbf{t}(\mathbf{T},\mathbf{T})}$, $\mathcal{I}_2 = 1, \mathcal{I}_3 = \lambda^{\mathbf{X}}_{\mathbf{t}(\mathbf{T},\mathbf{T})} = 1$. So only a pure time-reversal anomaly is present for eTmT when $\mathbf{X}$ permutes $e$ and $m$.

- Consider the case where $\mathbf{T}$ permutes $e$ and $m$, but $\mathbf{X}$ does not. Consistency requires that $\eta_\psi(\mathbf{T},\mathbf{T}) = \eta_\psi(\mathbf{T}',\mathbf{T}') = -1$. We can further gauge fix $\mathbf{t}(\mathbf{T},\mathbf{T}) = \mathbf{t}(\mathbf{T}',\mathbf{T}') = 1$. $\mathbf{t}(\mathbf{X},\mathbf{X})$

must be **T**-invariant, which follows from Eq. (111). In this case, $\mathcal{I}_1$ and $\mathcal{I}_2$ obviously vanish and we find $\mathcal{I}_3 = \theta_{t(\mathbf{X},\mathbf{X})}$, so the only anomalous one is $t(\mathbf{X},\mathbf{X}) = \psi$. This implies that the eCmC state, where $e$ and $m$ carry half $\mathbb{Z}_2$ charge, has a mixed anomaly when **T** permutes $e$ and $m$. This is the same anomaly structure as the case where **T** acts trivially.

- Consider the case where both **X** and **T** permute anyons. This is identical to the case above as long as we swap **T** and **T**'.

# Acknowledgments

We thank Zhenghan Wang, Cesar Galindo, and Xie Chen for helpful discussions. MB is supported by NSF CAREER (DMR-1753240), an Alfred P. Sloan Research Fellowship, and JQI-PFC-UMD. MC is supported by NSF CAREER (DMR-1846109).

# A  $F$, $R$, and $U$ symbols for Abelian Chern-Simons theories

Consider an Abelian Chern-Simons theory defined by a $D \times D$ $K$-matrix. Quasiparticles are labeled by charges under the U(1) gauge fields, each as a $D$-dimensional integer column vector $\mathbf{l} \in \mathbb{Z}^D$. Superselection sectors are defined by the equivalence relation $\mathbf{l} \sim \mathbf{l} + K\mathbf{n}$ where $\mathbf{n} \in \mathbb{Z}^D$. They form an Abelian group $\mathcal{A}$. For each anyon type $a$, we choose a representative charge vector, denoted by $\mathbf{l}_a$. Then $[\mathbf{l}_a]$ denotes the equivalence class associated with the representative $\mathbf{l}_a$. Clearly $[\mathbf{l}_{a\times b}] = [\mathbf{l}_a + \mathbf{l}_b]$. The topological twist factor is given by

$$\theta_a = e^{i\pi \mathbf{l}_a^\mathsf{T} K^{-1} \mathbf{l}_a}. \tag{130}$$

Below we first write down the $F$ and $R$ symbols for the special case where all matrix elements of $K$ are even. (The more general case requires dealing with the Sylow 2-group of $\mathcal{A}$).

We write the $F$ symbol as

$$[F_{a\times b\times c}^{a,b,c}]_{a\times b, b\times c} = e^{2\pi i \omega(a,b,c)}. \tag{131}$$

Since all particles are Abelian, the pentagon equation reduces to a 3-cocycle equation for $F$. Defining

$$\omega(a,b,c) = \frac{1}{2}\mathbf{l}_a^\mathsf{T} K^{-1}(\mathbf{l}_b + \mathbf{l}_c - \mathbf{l}_{b\times c}), \tag{132}$$

we prove that $\omega$ indeed defines a 3-cocycle on $\mathcal{A}$:

$$\omega(a,b,c) + \omega(a, b \times c, d) + \omega(b,c,d) - \omega(a \times b, c, d) - \omega(a, b, c \times d)$$
$$= \frac{1}{2}(\mathbf{l}_a + \mathbf{l}_b - \mathbf{l}_{a\times b})^\mathsf{T} K^{-1}(\mathbf{l}_c + \mathbf{l}_d - \mathbf{l}_{c\times d}). \tag{133}$$

Notice that $\mathbf{l}_a + \mathbf{l}_b - \mathbf{l}_{a\times b}$ must be of the form $K\mathbf{n}_1$ for some $\mathbf{n}_1 \in \mathbb{Z}^D$, and similarly $\mathbf{l}_c + \mathbf{l}_d - \mathbf{l}_{c\times d} = K\mathbf{n}_2$, so the result is $\frac{1}{2}\mathbf{n}_1^\mathsf{T} K\mathbf{n}_2$. Since we assume $K$ is even entry-wise, $\frac{1}{2}\mathbf{n}_1^\mathsf{T} K\mathbf{n}_2$ is an integer.

Now we define the $R$ symbol:

$$R_{a\times b}^{a,b} = e^{2\pi i r(a,b)}, \ \ r(a,b) = \frac{1}{2}\mathbf{l}_a^\mathsf{T} K^{-1} \mathbf{l}_b. \tag{134}$$

Suppose that now we have a symmetry group $G$ of the K-matrix, namely a set of invertible matrices $W_{\mathbf{g}}$ such that

$$W_{\mathbf{g}}KW_{\mathbf{g}}^{\mathsf{T}} = \sigma(\mathbf{g})K,$$
$$W_{\mathbf{g}}W_{\mathbf{h}} = W_{\mathbf{gh}}, \tag{135}$$

where $\sigma(\mathbf{g}) = \pm 1$ depending on whether $\mathbf{g}$ is space-time orientation reversing. Under $W_{\mathbf{g}}$ a charge vector $\mathbf{l}$ becomes $W_{\mathbf{g}}\mathbf{l}$, which induces an automorphism $\rho_{\mathbf{g}}$ on $\mathcal{A}$. Now we compute the $U$ symbol for this action. We have

$$\mathbf{l}_{\mathbf{g}a} = W_{\mathbf{g}}\mathbf{l}_a + K\mathbf{p}_{\mathbf{g},a}, \tag{136}$$

for $\mathbf{p}_{\mathbf{g},a} \in \mathbb{Z}^D$. It then follows that

$$\omega(\,^{\mathbf{g}}a, \,^{\mathbf{g}}b, \,^{\mathbf{g}}c)$$
$$= \frac{1}{2}\mathbf{l}_a^{\mathsf{T}}W_{\mathbf{g}}^{\mathsf{T}}K^{-1}W_{\mathbf{g}}(\mathbf{l}_b + \mathbf{l}_c - \mathbf{l}_{b\times c}) + \frac{1}{2}\mathbf{l}_a^{\mathsf{T}}W_{\mathbf{g}}(\mathbf{p}_{\mathbf{g},b} + \mathbf{p}_{\mathbf{g},c} - \mathbf{p}_{\mathbf{g},b\times c}) + \frac{1}{2}\mathbf{p}_{\mathbf{g},a}^{\mathsf{T}}W_{\mathbf{g}}(\mathbf{l}_b + \mathbf{l}_c - \mathbf{l}_{b\times c})$$
$$= \sigma(\mathbf{g})\omega(a,b,c) + \frac{1}{2}\mathbf{l}_a^{\mathsf{T}}W_{\mathbf{g}}(\mathbf{p}_{\mathbf{g},b} + \mathbf{p}_{\mathbf{g},c} - \mathbf{p}_{\mathbf{g},b\times c}), \tag{137}$$

where we have dropped a term $\frac{1}{2}\mathbf{p}_{\mathbf{g},a}^{\mathsf{T}}K(\mathbf{p}_{\mathbf{g},b} + \mathbf{p}_{\mathbf{g},c} - \mathbf{p}_{\mathbf{g},b\times c})$ as it contributes an integer multiple of $2\pi$. We have also dropped a term $\frac{1}{2}\mathbf{p}_{\mathbf{g},a}^{\mathsf{T}}W_{\mathbf{g}}(\mathbf{l}_b + \mathbf{l}_c - \mathbf{l}_{b\times c}) = \frac{1}{2}\mathbf{p}_{\mathbf{g},a}^{\mathsf{T}}W_{\mathbf{g}}K\mathbf{n} \in \mathbb{Z}$, because $K$ is even entry-wise, and thus this term also contributes an integer multiple of $2\pi$. In going from the first to the second line we have also used the following

$$W_{\mathbf{g}}^{\mathsf{T}}K^{-1}W_{\mathbf{g}} = W_{\mathbf{g}}^{\mathsf{T}} \cdot \sigma(\mathbf{g})(W_{\mathbf{g}}^{\mathsf{T}})^{-1}K^{-1}W_{\mathbf{g}}^{-1} \cdot W_{\mathbf{g}} = \sigma(\mathbf{g})K^{-1}. \tag{138}$$

Considering the group multiplications, we find

$$\mathbf{p}_{\mathbf{gh},a} = \sigma(\mathbf{g})(W_{\mathbf{g}}^{-1})^{\mathsf{T}}\mathbf{p}_{\mathbf{h},a} + \mathbf{p}_{\mathbf{g},\,^{\mathbf{h}}a}. \tag{139}$$

Now let us specialize to the case studied in Sec. 8.5.5, with the K-matrix given in Eq. 102 and the $\mathbb{Z}_4^{\mathbf{T}}$ symmetry given in Eq. 103. In this case $\sigma(\mathbf{g}) = q(\mathbf{g})$, where recall $q(\mathbf{g}) = \pm 1$ depending on whether $\mathbf{g}$ corresponds to a unitary or anti-unitary symmetry. Also, in this case we have $W_{\mathbf{g}}^T = q(\mathbf{g})W_{\mathbf{g}}$.

We thus find that we can set

$$U_{\mathbf{g}}(\,^{\mathbf{g}}a, \,^{\mathbf{g}}b) = \exp(-i\pi\mathbf{l}_a^{\mathsf{T}}W_{\mathbf{g}}p_{\mathbf{g},b}). \tag{140}$$

$\kappa$ is defined as:

$$\kappa_{\mathbf{g},\mathbf{h}}(a,b) = U_{\mathbf{gh}}(a,b)[U_{\mathbf{g}}(a,b)]^{-1}[U_{\mathbf{h}}(\,^{\bar{\mathbf{g}}}a, \,^{\bar{\mathbf{g}}}b)]^{-q(\mathbf{g})}. \tag{141}$$

Thus we find:

$$\kappa_{\mathbf{g},\mathbf{h}}(\,^{\mathbf{gh}}a, \,^{\mathbf{gh}}b) = \exp\left(-i\pi[\mathbf{l}_a^T W_{\mathbf{gh}}p_{\mathbf{gh},b} - l_{\mathbf{h}a}^T W_{\mathbf{g}}p_{\mathbf{g},\,^{\mathbf{h}}b} - q(\mathbf{g})l_a^T W_{\mathbf{h}}p_{\mathbf{h},b}]\right)$$
$$= \exp\left(-i\pi[q(\mathbf{g})\mathbf{l}_a^T(W_{\mathbf{g}}W_{\mathbf{h}}(W_{\mathbf{g}}^T)^{-1} - W_{\mathbf{h}})\mathbf{p}_{\mathbf{h},b} + \mathbf{l}_a^T(W_{\mathbf{g}}W_{\mathbf{h}} - W_{\mathbf{h}}^T W_{\mathbf{g}})p_{\mathbf{g},\,^{\mathbf{h}}b}]\right). \tag{142}$$

Using $W_{\mathbf{g}}^T = q(\mathbf{g})W_{\mathbf{g}}$ and $W_{\mathbf{gh}} = W_{\mathbf{hg}}$ for this example, we see that $W_{\mathbf{g}}W_{\mathbf{h}}[W_{\mathbf{g}}^T]^{-1} - W_{\mathbf{h}} = q(\mathbf{g})W_{\mathbf{h}} - W_{\mathbf{h}}$ and $W_{\mathbf{g}}W_{\mathbf{h}} - W_{\mathbf{h}}^T W_{\mathbf{g}} = W_{\mathbf{gh}}(1 - q(\mathbf{h}))$. Since $q(\mathbf{g}) = \pm 1$, it follows that the entries in the brackets in the second line of Eq. (142) are all even, so that $\kappa_{\mathbf{g},\mathbf{h}}(a,b) = 1$ for all choices of $a, b, \mathbf{g}, \mathbf{h}$.

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
