# Peer review of "Relative Anomalies in (2+1)D Symmetry Enriched Topological States"

_SciPost Physics, doi:SciPost Phys. 8, 028 (2020)_

## Round 1 · Referee Report · Anonymous (Referee 1) · 2019-9-9

Strengths

1- Not only their new results, it also contains a concise and readable summary of the framework, including why the symmetry fractionalization classes are a torsor over $H^2_{\rho}(G,\mathcal{A})$.

Weaknesses

1- The authors gave the formula for the relative anomaly (43) but did not show that it is actually a cocycle. (The referee understands that the authors mentioned that "however we do not pursue this further here". But it is a weakness.)

Report

In this paper, the authors obtained a universal formula for the change of the anomaly valued in $H^4(G,U(1))$ of a 2+1d TQFT under the shift of the symmetry fractionalization class by an element of $H^2_{\rho}(G,\mathcal{A})$. This universal formula was then applied to many concrete cases, reproducing many results previously obtained in other papers by a case-by-case analysis.

The referee found the paper clearly written and containing interesting results, thus worth publishing on SciPost.

The referee has one question and one suggestion:

1- The relative anomaly is (more or less) a map from $ H^2_{\rho}(G,\mathcal{A})$ to $H^4(G,U(1))$. What is it? $M_{ab}$ provides a bilinear map $M:\mathcal{A}\times\mathcal{A}\to U(1)$, so there is a natural quadratic pairing $M(t,s)\in H^4(G,U(1))$, given $t,s\in H^2_{\rho}(G,\mathcal{A})$. Maybe $I(t)$ is a quadratic refinement of $M(t,s)$, in the sense $I(t+s)=I(t)I(s)M(t,s)$?

2- The authors might want to comment on https://arxiv.org/abs/1805.02738 in your section VI. There, the $\mathbb{Z}_2^T$ anomalies of abelian anyon systems were extensively studied. Your $H^2_{\rho}(\mathbb{Z}_2^T,\mathcal{A})$ was denoted by $C=\mathrm{Ker}(1-T)/\mathrm{Im}(1+T)$, and the total anomaly was identified as $\mathrm{Arf}(q)$ where $q(a)=\theta(a)\eta(a)$ is considered as a function on $C$. Since $\mathrm{Arf}(q)$ has the well-known property $\mathrm{Arf}(tq)=q(t) \mathrm{Arf}(q)$, this implies your relative anomaly formula. Also, in this case at least, $q(t+s)=q(t)q(s) M(t,s)$.

Requested changes

All the requested changes are extremely minor typos:

The fist line of I. Introduction:

The last few years ... has seen major progress $\to$ The last few years ... have seen major progress

some lines below it:

an important class of invertible states are $\to$ an important class of invertible states is

(22):

$|a,b;c,\nu\rangle$ on the RHS should probably be $|^ga,{}^gb;{}^gc,\nu\rangle$

some lines above (29):

Eq. (28) should be Eq.~(28) in the LaTeX source file; a period following a small-case letter is known to automatically produce a wider space, and this feature needs to be suppressed here.

two lines above (65):

a (absolute) vison $\to$ an (absolute) vison

The first line of VIII.B:

Sec. II, III should be Sec.~II, III

three lines above (110):

We now given an ... $\to$ We now give an ...

one line above (119):

The sentence "The $\mathbb{Z}_2$ charge conjugation symmetry $C:a\to N-a$." lacks a verb.

In (125):

The first line there are $R^{ta} R^{at}$ which are converted to $M_{ta}$ in the second line. The second line also contains $M_{tb}$ but there is no corresponding $R^{tb}R^{bt}$ in the first line.

---

## Round 2 · Author Response

Regarding the referee's first question, indeed this was discussed in the paper by Etingof, Nikshych, and Ostrik (see https://arxiv.org/pdf/0909.3140.pdf, Proposition 8.12 ). The formulas derived by ENO apply to the case where the symmetry actions do not permute the anyons. The generalization to the case where anyons are permuted has not been proven, as far as we are aware.
Regarding the referee's second comment, we have included the discussion of the Abelian case summarized by the referee at the end of Sec. VI of our revised version.
Finally, we have made the additional grammatical changes suggested by the referee.

---

## Round 2 · List of Changes

1. Added a short discussion on the relation between the relative anomaly formula for Z2^T symmetry for Abelian anyons and the arf invariant discussed in the paper of Lee and Tachikawa (2018) .
2. Fixed a typo in Table I and clarified the theories it applies to.
3. Fixed some minor typos and grammatical issues mentioned by the referee.

---

## Editorial Decision

published